# Epstein-Barr virus protein EBNA-LP engages YY1 through leucine-rich motifs to promote naïve B cell transformation

**Jana M. Cable[1], Nicolás M. Reinoso-Vizcaino[1], Robert E. White[2], Micah A. Luftig[1]***

**1** Department of Molecular Genetics and Microbiology, Duke University School of Medicine, Durham, North Carolina, United States of America, **2** Section of Virology, Department of Infectious Disease, Imperial College London, London, United Kingdom

* micah.luftig@duke.edu

**Data Availability Statement:** RNA-Seq and Cut&Run data sets are publicly available in the NIH Gene Expression Omnibus (GEO) with the following accession numbers: RNA-seq data from

## Abstract

Epstein-Barr Virus (EBV) is associated with numerous cancers including B cell lymphomas. *In vitro*, EBV transforms primary B cells into immortalized Lymphoblastoid Cell Lines (LCLs) which serves as a model to study the role of viral proteins in EBV malignancies. EBV induced cellular transformation is driven by viral proteins including EBV-Nuclear Antigens (EBNAs). EBNA-LP is important for the transformation of naïve but not memory B cells. While EBNA-LP was thought to promote gene activation by EBNA2, EBNA-LP Knockout (LPKO) virus-infected cells express EBNA2-activated cellular genes efficiently. Therefore, a gap in knowledge exists as to what roles EBNA-LP plays in naïve B cell transformation. We developed a trans-complementation assay wherein transfection with wild-type EBNA-LP rescues the transformation of peripheral blood- and cord blood-derived naïve B cells by LPKO virus. Despite EBNA-LP phosphorylation sites being important in EBNA2 co-activation; neither phospho-mutant nor phospho-mimetic EBNA-LP was defective in rescuing naïve B cell outgrowth. However, we identified conserved leucine-rich motifs in EBNA-LP that were required for transformation of adult naïve and cord blood B cells. Because cellular PPAR-g coactivator (PGC) proteins use leucine-rich motifs to engage transcription factors including YY1, a key regulator of DNA looping and metabolism, we examined the role of EBNA-LP in engaging transcription factors. We found a significant overlap between EBNA-LP and YY1 in ChIP-Seq data. By Cut&Run, YY1 peaks unique to WT compared to LPKO LCLs occur at more highly expressed genes. Moreover, Cas9 knockout of YY1 in primary B cells prior to EBV infection indicated YY1 to be important for EBV-mediated transformation. We confirmed EBNA-LP and YY1 biochemical association in LCLs by endogenous co-immunoprecipitation and found that the EBNA-LP leucine-rich motifs were required for YY1 interaction in LCLs. We propose that EBNA-LP engages YY1 through conserved leucine-rich motifs to promote EBV transformation of naïve B cells.

WT and LPKO LCLs (GSE268054), YY1 Cut&Run in WT and LPKO LCLs (GSE268305), and YY1 Cut&Run in LPKO LCLs reconstituted with wild type EBNA-LP, LRM Mutant, or empty vector (GSE268056).

**Funding:** This work was funded by NIH grant R01CA140337 (to M.A.L.), T32CA009111 (M.A.L.), and F31DE031509 (to J.M.C.). The funders had no role in study design, data collection and analysis, decision to publish, or preparation of the manuscript.

**Competing interests:** The authors have declared that no competing interests exist.

## Author summary

Epstein-Barr Virus (EBV) is associated with various B cell lymphomas, particularly in immunosuppressed individuals. In the absence of a functional immune system, viral latency proteins, including EBV Nuclear Antigens (EBNAs) act as oncoproteins to promote tumorigenesis. EBNA-LP is one of the first viral proteins produced after infection and is important for the transformation of naïve B cells. However, the roles of EBNA-LP during infection are largely undefined. In this study, developed an assay in which the role of wild type and mutant EBNA-LP could be investigated in the context of primary naïve B cells infected with an EBNA-LP Knockout virus. Using this assay, we identified highly conserved leucine-rich motifs within EBNA-LP that are important for transformation of EBV-infected naïve B cells. These conserved motifs associate with the cellular transcription factor YY1, an important transcriptional regulator in B cell development and in many cancers, that we now show is essential for outgrowth of EBV infected B cells. Our study provides further insights into the mechanisms by which EBV transforms naïve B cells.

## Introduction

Epstein-Barr virus (EBV) is a gamma-herpesvirus that infects–and permanently resides in–nearly all individuals by adulthood. EBV infects resting B cells, which leads to B cell activation followed by a life-long latent infection in memory B cells [1]. While infection is typically asymptomatic, EBV is associated with numerous malignancies including B cell lymphomas such as Hodgkin's Lymphoma, Burkitt Lymphoma, and lymphoproliferative disease in immunocompromised individuals [2]. EBV-induced cellular proliferation is largely driven by the activity of viral proteins.

EBV infection of B cells leads to the expression of the viral latency program which includes EBV Nuclear Antigens (EBNAs) and Latent Membrane Protein (LMPs). EBNAs are transcriptional regulators which include EBNA1, 2, 3A, 3B, 3C, and -Leader Protein (LP). Other than EBNA1, the EBNA proteins do not directly bind DNA, but rather regulate transcription by engaging with cellular DNA sequence-specific binding proteins. EBNA2 and EBNA-LP are the initial viral latency proteins expressed, being detected as early as 12 hours post infection from the repeated viral W promoter (Wp) [3–5]. Once EBNA2 and EBNA-LP are expressed, the C promoter (Cp) becomes transcriptionally active from which promoter EBNAs are then predominantly transcribed, followed by the LMP proteins which take 1–2 weeks to reach the level sustained in LCLs [5,6]. *In vitro*, infected cells initially increase in size, followed by a phase of cellular hyperproliferation beginning 3–4 days post infection, followed by a slower period of cellular outgrowth [7]. Restriction by innate cellular barriers including metabolic stress and DNA damage ultimately leads to cellular arrest of many infected cells [8]. However, cells that overcome these barriers *in vitro* are transformed into immortalized lymphoblastoid cell lines (LCLs), serving as a model to study the role of viral proteins in primary B cell infection and EBV-associated malignancies.

EBNA2 is an important regulator of both viral and cellular genes and is required at all stages of primary infection *in vitro* [9]. Importantly, EBNA2 engages several host transcription factors including RBPJ and EBF1 to induce high levels of the oncogene Myc, among other cellular genes [10–12]. While less is known about the role of EBNA-LP in EBV infection and cellular transformation, several studies have contributed to our knowledge of EBNA-LP activity in infected B cells. The main function of EBNA-LP has been ascribed to co-activating EBNA2

activity at both viral and host genes. In transfection experiments performed outside of the context of primary infection, EBNA-LP increases EBNA2 activation of the Cp, LMP1 promoter, RBPJ-binding sites, and cellular genes including HES1 [13–15]. However, studies identifying which domains of EBNA-LP are responsible for this activity have found conflicting results [13–15]. EBNA-LP is also hypothesized to promote EBNA2 activity by removing transcriptional repressors including NCoR and HA95 from EBNA2 targeted sites [15–17]. EBNA-LP enriched sites on chromatin are also associated with several cellular transcription factors and DNA looping factors [15–18]. Furthermore, upon B cell infection and in LCLs, EBNA-LP localizes to the subnuclear foci called PML bodies, which are implicated in restricting viral gene expression [19,20]. However, the significance of these many EBNA-LP functions in the context of primary infection are still unclear.

The N-terminal part of EBNA-LP is composed of variable numbers of a repeated 66 amino acid domain (encoded by W1 and W2 exons within the major internal repeat of the EBV genome, which we will call the W domain), and its C-terminus (called the Y domain, as it derives from exons Y1 and Y2) contains 45 amino acids. Circulating viruses encode at least 4 repeated W domains [21]. Alternative Wp usage and splicing leads to expression of multiple EBNA-LP isoforms with variable numbers of W domains. Across the W and Y domains, there are five evolutionarily conserved regions (CR) between EBV EBNA-LP and its homologs in primate lymphocryptovirus (LCV) [22]. One study found that all three conserved regions in the W domain (CR1, CR2, CR3) are required to enhance the activation of genes by EBNA2, while motifs within CR1 and CR2 contain a bi-partite nuclear localization signal [22–24]. The Y domain contains two additional conserved regions (CR4 and CR5) [22]. Importantly, the significance of the conserved regions has not been studied in the context of primary infection.

In addition to these conserved regions, EBNA-LP also contains several phosphorylated serine residues in the W domain. One of which, serine 36 (S36), is highly conserved across EBNA-LP homologs in lymphocryptovirus [23,25–27]. Serine 34 (S34) and 63 (S63), in CR3, are also possible sites of phosphorylation, although to date modification at S34 and S63 has not been validated [24]. EBNA-LP can be phosphorylated in a cell cycle-dependent manner by numerous kinases including Casein Kinase II, p34$^{cdc2}$, DNA-PK, and the viral kinase BGLF4, with EBNA-LP being most highly phosphorylated during $G_2$ [24,27–30]. In transfection assays, phosphorylation of EBNA-LP is required for its ability to help EBNA2 to induce Cp and LMP1 transcription [23,24,29]. Phosphorylation of EBNA-LP is also required for its association with the histone acetyltransferase p300 [31].

EBNA-LP is essential in transforming cord blood-derived B cells which are exclusively naïve, and is significantly more important for transformation of sorted adult peripheral blood naïve B cells, defined as IgD+/CD27-, than adult peripheral blood memory B cells (CD27+) [32]. A similar study in adenoid tissue-derived naïve B cells found EBNA-LP was critical during early stages of infection for cell division and survival [9]. In the absence of the EBNA-LP Y domain, the efficiency of cellular transformation of infected B cells is significantly reduced [33,34]. While the function of the Y domain is unknown, leucines in the Y domain CR4 have been implicated as the site of interaction with the transcription factor Estrogen Related Receptor Alpha (ERR$\alpha$) [35]. Intriguingly, ERR$\alpha$ is a transcription factor that is co-activated by the Peroxisome PPAR-g Coactivator (PGC) family of proteins [36]. The PGC proteins, like many other transcriptional co-activators, utilize their leucine-rich motifs (LRMs) or nuclear receptor box motifs typically of the sequence LXXLL to bind transcription factors and then recruit additional co-activators to promote transcription [36–39]. We therefore sought to determine whether the LRMs within EBNA-LP could be important motifs functionally mimicking the LRMs in cellular PGC co-activators of transcription.

Despite the proposed role of EBNA-LP in assisting EBNA2-mediated transcriptional activation, genetic studies with an EBNA-LP Knockout (LPKO) virus indicates EBNA-LP is in fact not required for EBNA2 to activate host genes including Myc, nor for EBNA2 recruitment to cellular chromatin [32], but rather may constrain excessive EBNA2-mediated activation of cellular genes. Therefore, we sought to uncover other important roles of EBNA-LP in EBV-infected B cells in a physiologically relevant context. To do so, we used a trans-complementation assay to define the importance of conserved regions and posttranslational modifications of EBNA-LP in the transformation of naïve B cells by EBV.

## Results

### Naïve B Cells Infected with LPKO Virus are Rescued by Trans-Complementation with Wild-Type EBNA-LP

In order to assay the consequences of EBNA-LP mutations, we developed a trans-complementation assay. First, we tested whether outgrowth of adult naïve B cells infected with EBNA-LP Knockout (LPKO) virus could be rescued by expression of exogenous, wild type EBNA-LP. We created a complementation vector encoding FLAG-tdTomato, a P2A cleavage site, wild type FLAG-EBNA-LP including four of the repeated W domains codon optimized to improve plasmid stability (S1 Table), and the EBV oriP and EBNA1 to allow episomal persistence of the vector (S1A and S1B Fig). Naïve B cells derived from adult peripheral blood mononuclear cells (PBMCs) were isolated by immunomagnetic selection for IgD+/CD27- cells, although purity varied by donor ranging from 92.2–97.6% with the majority of contaminating cells coming from IgD-/CD27- B cells, highlighting the challenges of isolating pure naïve B cell from adult PBMCs (S2A Fig). For each replicate, isolated cells were transfected with the complementation vector encoding EBNA-LP, with vector encoding only FLAG-tdTomato, or left untransfected (Fig 1A). Cells were then infected with LPKO or WT virus [32] at a titer determined to infect almost 100% of cells. Outgrowth of tdTomato positive cells, indicative of maintenance of the episomal vector, was then assayed by flow cytometry every seven days post transfection, with compensation for GFP produced by the virus (S2B Fig). While the electroporation of primary B cells resulted in killing of some transfected cells, the similarity in total proliferating cells between untransfected cells and cells transfected with only vector for WT-infected cells indicated that the failure of LPKO-infected cells transfected with vector to proliferate was not a result of electroporation, but rather reflects the importance of EBNA-LP early during infection in peripheral blood-derived naïve B cells (S2C and S2D Fig).

In all three adult blood donors, trans-complementation of naïve B cells with EBNA-LP that were infected with LPKO virus led to outgrowth of tdTomato positive cells over time similar to WT virus infected cells transfected with the tdTomato vector lacking EBNA-LP (Fig 1B). Of note, there was a high degree of variability in outgrowth efficiency between replicates both within and between donors. LPKO infected cells transfected with vector encoding only tdTomato consistently failed to expand tdTomato positive cells, whereas trans-complementation with EBNA-LP rescued outgrowth and successfully generated tdTomato+ LCLs (Fig 1C–1E). As expected, WT virus transformed untransfected naïve B cells into tdTomato negative LCLs in each replicate across three donors (Fig 1E). WT infected cells transfected with vector also generated LCLs and retained tdTomato expression, albeit only in a small proportion–on average fewer than 3%–of cells (S2E Fig). As the purified naïve B cell populations still contain contaminating unconventional memory B cells (IgD-/CD27-) (S2A Fig), which do not require EBNA-LP for outgrowth and cellular transformation [32,40], tdTomato negative LCLs were observed following LPKO infection in both untransfected and cells trans-complemented with only tdTomato in some donors (Figs 1E and S2E). Rescued LCLs retained EBNA-LP

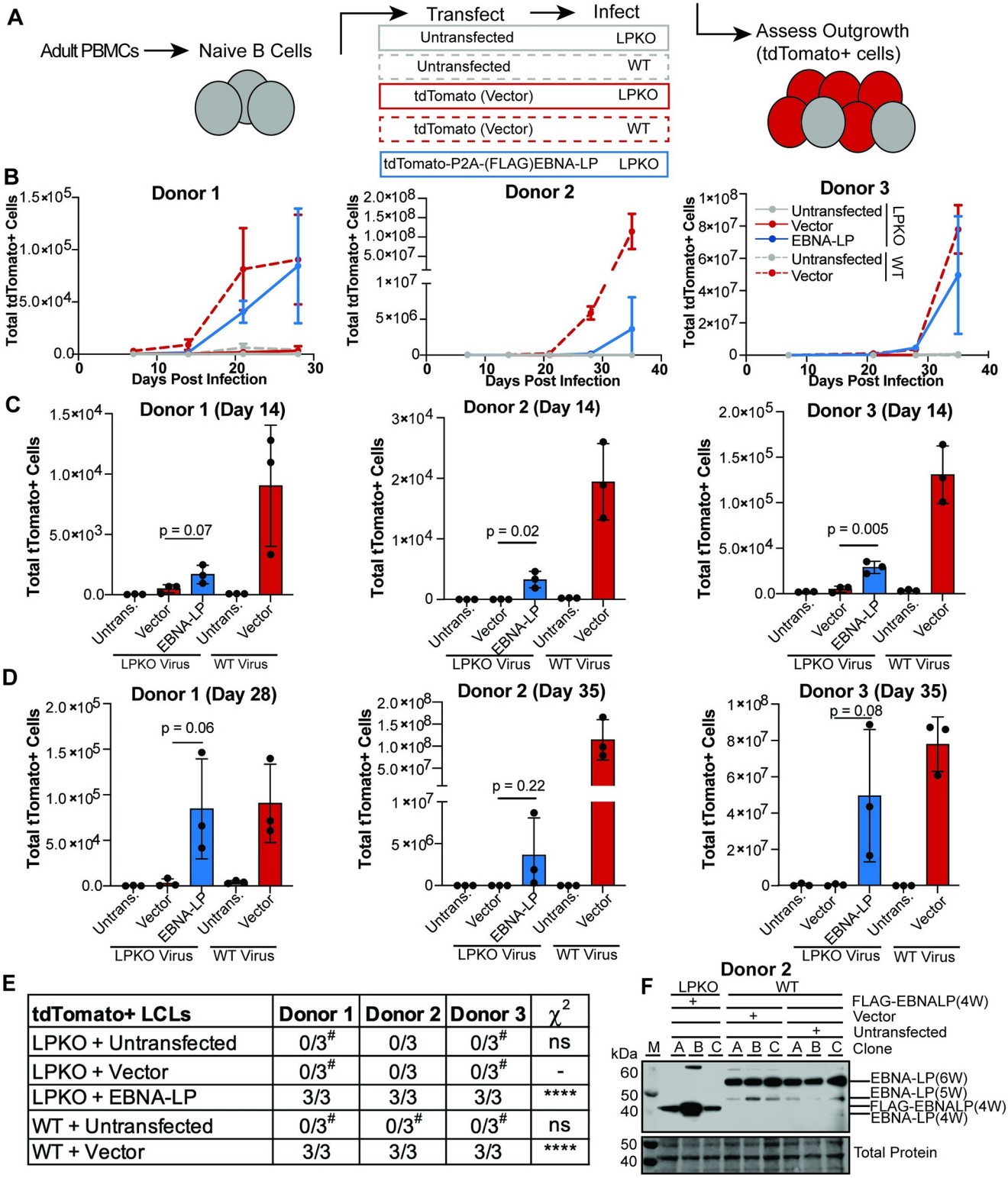

**Fig 1. Trans-complementation rescues LPKO virus infected naïve B cells. A**. Schematic of experimental design. **B.** Outgrowth of tdTomato positive cells over time (n = 3). WT infected cells are indicated with dashed lines, while LPKO infected cells are solid lines. **C**. Total tdTomato positive cells in each condition at 14 days post-infection (n = 3). P values from unpaired t-test. Mean and standard deviation are plotted. **D**. Total tdTomato positive cells at the final time point for each donor, 28 or 35 days post-infection (n = 3). **E**. Number of tdTomato positive LCLs generated per condition. #Indicates conditions in which tdTomato negative LCLs were generated. P values calculated using Fisher's exact test to compare transformation outcomes to LPKO + Vector. ****

indicates a p value < 0.0001. **F**. EBNA-LP expression in trans-complemented LPKO LCLs and WT LCLs. W indicates number of W domains in EBNA-LP protein expressed from virus or expression construct. M indicates molecular weight marker.

expression at levels comparable to WT LCLs (Fig 1F). Notably, outgrowth of tdTomato positive LPKO infected memory B cells was not impacted by transfection with wild type EBNA-LP compared to vector (S3A–S3C Fig). The ability of this trans-complementation assay to consistently rescue transformation of LPKO infected naïve B cells and retain EBNA-LP expression confirms this assay can also be used to assess the role of EBNA-LP mutants in the context of primary infection.

## Phosphorylation of EBNA-LP is not Required for Naïve B Cell Transformation

Phosphorylation of EBNA-LP is considered essential for EBNA-LP-mediated coactivation of EBNA2 [23,24,29]. Therefore, we sought to uncover whether phosphorylation is required for EBNA-LP-mediated transformation of naïve B cells. All three predicted phosphorylated serines (S34, S36, and S63) are conserved across EBV Type 1 and Type 2 strains, while S36 is also conserved across EBV strains and in strains of the non-human primate lymphocryptovirus (LCV), the closest common ancestor to EBV [22,24] (Fig 2A). We assessed phosphorylation of EBNA-LP by performing FLAG immunoprecipitation followed by mass spectrometry with trypsin digestion of our FLAG-tagged EBNA-LP encoding 4 W domains expressed in 293T cells. In total, 81 peptides were identified which covered most of the EBNA-LP protein (Fig 2B). Only two peptides spanning S34 and S36 were identified, likely because they are located in a region rich in arginine, the site of trypsin cleavage (Fig 2B) [41,42]. However, of these identified peptides, one contained phosphorylated S34 (32-HRSpPSPTR-39) (Figs 2C and S4A) and the other phosphorylated S36, (34-SPSpPTRGGQEPR-45) suggesting phosphorylation is a highly abundant modification at S34 and S36 (Figs 2C and S4B). Peptides spanning S63 were also identified (including 51-VLVQQEEEVVSGSpPSGPR-68) (S4C Fig) with phosphorylation at S63 observed in 6 out of 41 peptides (Fig 2C). While phosphorylation of S36 has previously been confirmed, this is the first study to validate S34 and S63 as phosphorylation sites on EBNA-LP by mass spectrometry.

We then generated both a FLAG-tagged phospho-mutant (S3A) in which S34, S36, and S63 were modified to alanine, and a phospho-mimetic (S3E) in which S34, S36, and S63 were modified to glutamic acid. Wild Type, S3A, and S3E EBNA-LP were transfected into 293T cells, and FLAG immunoprecipitations to enrich for EBNA-LP were performed in triplicate for each construct prior to analysis by mass spectrometry using Tandem Mass Tag (TMT) labelling to quantify the signal of identified peptides across samples. Only wild type EBNA-LP was phosphorylated, with peptides containing phospho-residue S63, but no phosphorylated peptides were detected from either EBNA-LP-S3A or -S3E (Fig 2D), although no peptides covering S34 and S36 were identified in this experiment. This confirms S34, S36, and S63 as the only sites of phosphorylation on EBNA-LP, further validating that other highly conserved serines including S61 are not phosphorylated.

We then tested whether EBNA-LP-S3A or -S3E could rescue LPKO virus infected naïve B cells. Despite the previously inferred importance of EBNA-LP phosphorylation sites, we found that, surprisingly, trans-complementation with either the phospho-mutant or phospho-mimetic could rescue LPKO infected naïve B cells to the same degree as wild-type EBNA-LP both 14 days post infection (Fig 2E) and 28–35 days post infection (Fig 2F). Both mutants also successfully generated tdTomato positive LCLs similar to wild type EBNA-LP (Fig 2G). Further, EBNA-LP-S3A

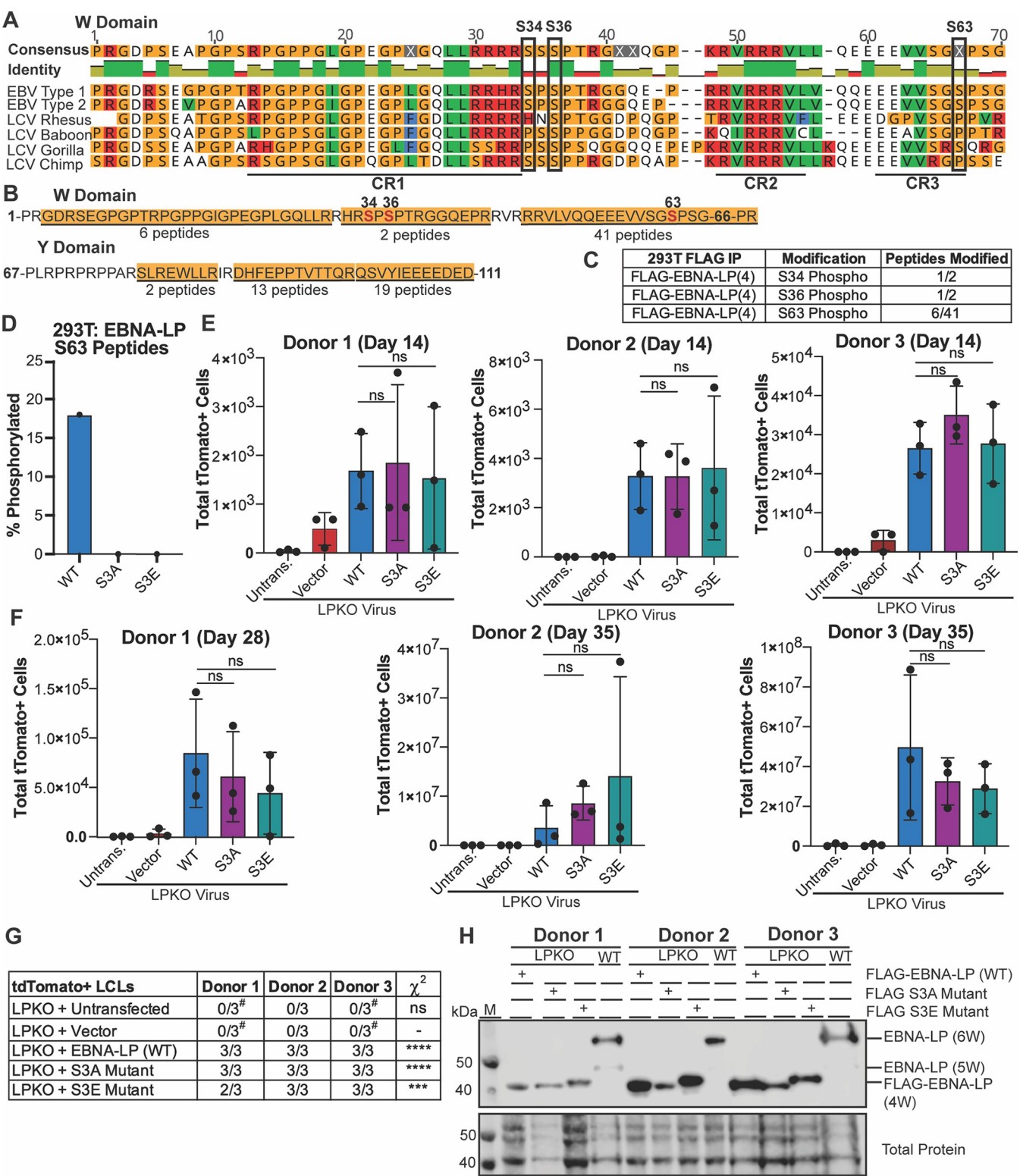

**Fig 2. Phosphorylation of EBNA-LP is not required for EBV-infected naïve B cell transformation. A**. Known phosphorylated serines in EBNA-LP W domain across human EBV strains (Type 1 is B95-8, Type 2 is AG876) and EBNA-LP homologs in primate LCV strains. Identity bar indicates conservation of amino acids, with green indicating highly conserved regions, yellow indicating less conserved, and red are poorly conserved regions. Colors of residues indicate amino acid properties. Phosphorylated serines are indicated by black boxes. Previously defined conserved regions (CR1, CR2, CR3) are underlined (22). **B**. Amino acid sequence coverage of FLAG-tagged EBNA-LP (wild type) construct encoding 4 W domains expressed in 293T cells, followed by mass

spectrometry of FLAG immunoprecipitation. Underlined and highlighted residues indicate peptides identified by mass spectrometry, with the number of peptides indicated below. Phosphorylated residues are red, with residue number above. **C.** Abundance of phosphorylated peptides for all identified phospho-serines out of the total peptides spanning that residue from FLAG immunoprecipitation in B. **D.** Average percent of total Tandem Mass Tag (TMT) signal for peptides with S63 phosphorylation when FLAG-tagged EBNA-LP (wild type), the EBNA-LP S3A mutant, and EBNA-LP S3E mutant are expressed in 293T cells prior to FLAG immunoprecipitation, TMT labeling, and mass spectrometry (n = 3). **E.** Total tdTomato positive cells at 14 days post infection for each donor in LPKO virus infected, trans-complemented cells (n = 3). Significance determined by unpaired t-test. Mean and standard deviation are plotted. **F.** Total tdTomato positive cells at 28 or 35 days weeks post infection for each donor (n = 3). **G.** Total tdTomato+ LCLs generated per condition. #Indicates conditions in which tdTomato negative LCLs were generated. P values calculated using Fisher's exact test to compare transformation outcomes to LPKO + Vector. **** indicates a p value < 0.0001. *** indicates a p value < 0.001. **H.** Trans-complemented LPKO LCLs express wild type and mutant FLAG-tagged EBNA-LP. W indicates number of W domains in EBNA-LP protein expressed from virus or expression construct. M indicates molecular weight marker.

and -S3E constructs were retained and expressed EBNA-LP protein in trans-complemented LCLs (Fig 2H). These findings suggest that the EBNA2 co-activation assay may not faithfully recapitulate the functions important for EBNA-LP in naïve B cell transformation.

## EBNA-LP Contains Conserved, Leucine-Rich Motifs that are Essential for Naïve B Cell Transformation

Since the sites of EBNA-LP phosphorylation are non-essential for transformation of naïve B cells, we next sought to determine what other motifs in EBNA-LP could be important for mediating naïve B cell transformation. We identified highly conserved leucine-rich motifs (LRMs) in both the repeated W domain CR1s and Y domain CR4 [22,35] (Fig 3A). Because these motifs are reminiscent of the leucine-rich motifs in the PGC coactivator family, we hypothesized that these conserved regions could be important for mediating EBNA-LP interaction with cellular factors. We therefore generated constructs in which the leucines in LRMs of both the W domain CR1s and Y domain CR4 were mutated to alanine (LRM mutant) (Fig 3B). At 14 days post infection, there were significantly fewer tdTomato positive cells in the LPKO virus infected cells trans-complemented with the LRM mutant compared to trans-complementation with wild type EBNA-LP (Fig 3C). At this early time point there was also a higher total number of cells in LPKO virus rescued by wild type EBNA-LP compared to the LRM mutant (S5A Fig), further supporting the inability of the LRM mutant to rescue LPKO virus early during infection. Trans-complementation with wild type EBNA-LP continued to promote higher outgrowth of tdTomato positive cells compared to the LRM mutant 28–35 days post infection (Fig 3D), and the LRM mutant failed to generate tdTomato positive LCLs from LPKO infected adult naïve B cells (Fig 3E). We did, however, observe rare outgrowth of tdTomato negative LCLs in all donors as even a small level of contaminating memory cells could ultimately take over the culture during weeks of B cell outgrowth (Figs 3E and S5B and S5C).

As cord blood provides a purer naïve B cell population [43], we next assessed the ability of the LRM mutant to rescue LPKO infected cord blood B cells. Like adult naïve B cells, the cord blood B cells could support outgrowth of rescued, tdTomato-positive cells over time (Fig 4A and 4B) resulting in a higher total number of cells (S6A and S6B Fig) and establishment of tdTomato-positive LPKO LCLs when trans-complemented with wild type EBNA-LP but not with the LRM mutant (Figs 4C and S5B and S6D). These findings confirm that the conserved leucine-rich motifs are important for EBNA-LP activity in naïve B cells.

## Trans-Complemented EBNA-LP Leucine-Rich Mutant is Expressed and Localizes Similar to Wild Type EBNA-LP in LCLs Derived from Adult Total B Cell Infection

We next sought to confirm that the EBNA-LP LRM mutant had wild type level expression. While the LRM mutant was unable to generate tdTomato positive LCLs from infected naïve B

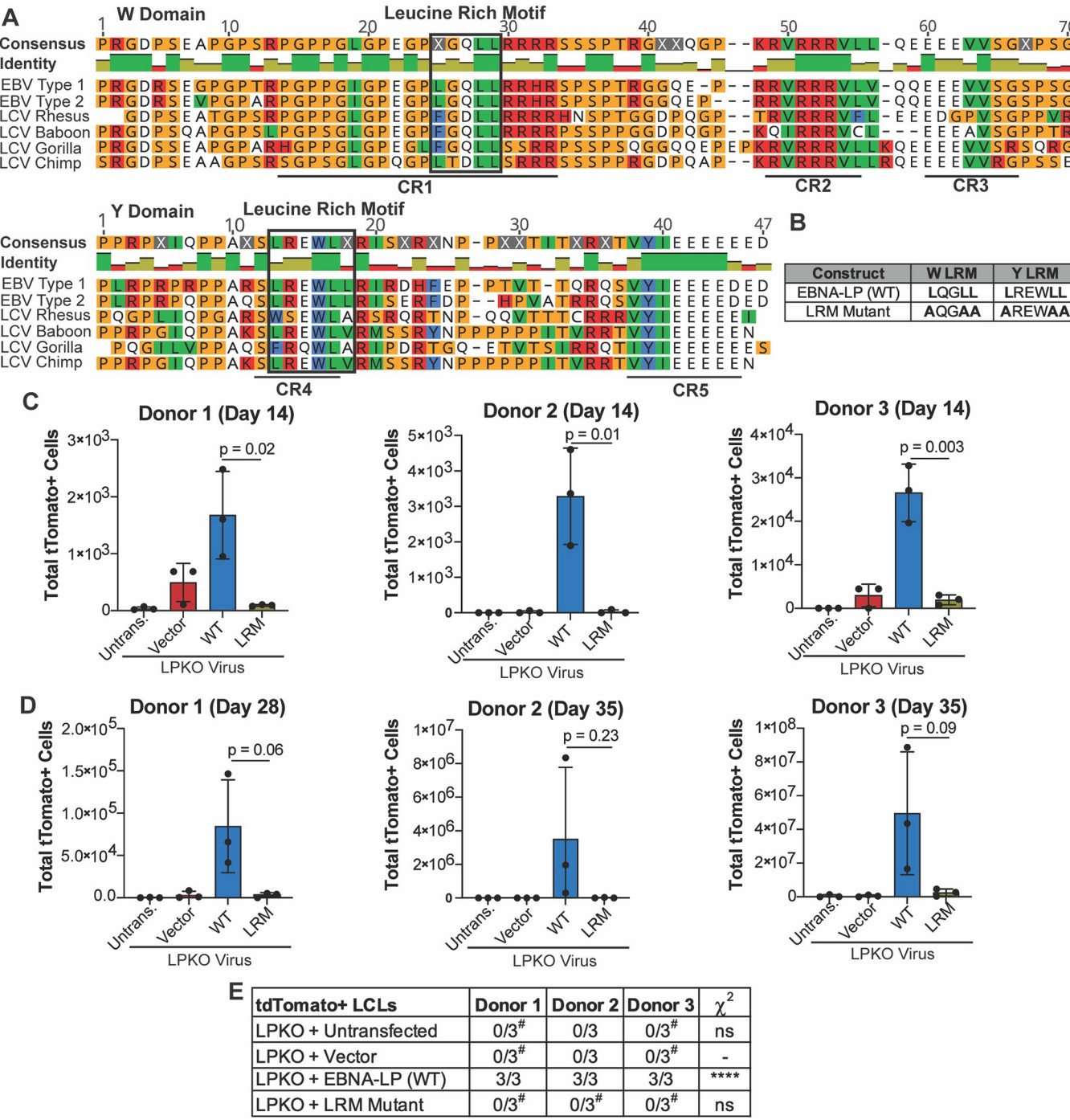

**Fig 3. EBNA-LP contains Leucine-rich motifs which are important for adult naïve B cell transformation. A**. Alignment of EBNA-LP W and Y domains across human EBV (Type 1 is B95-8, Type 2 is AG876) and LCV strains, with identified leucine-rich motifs highlighted in black boxes. Identity bar indicates conservation of amino acids, with green indicating highly conserved regions, yellow indicating less conserved, and red are poorly conserved regions. Colors of residues indicate amino acid properties. Previously defined conserved regions (CR1, CR2, CR3, CR4, CR5) are underlined (22). **B**. LRM Mutant construct generated with mutations in the leucine-rich domains has modified leucines to alanines. **C**. Total tdTomato positive cells at 14 days post infection for each donor in LPKO virus infected, trans-complemented cells (n = 3). **D.** Total tdTomato positive cells in adult naive B cells from three donors at 28 or 35 days post infection (n = 3). P values from unpaired t-test. Mean and standard deviation are plotted. **E.** Total tdTomato positive LCLs generated from trans-complemented LPKO infected adult naïve B cells. #Indicates conditions in which some replicates generated tdTomato negative LCLs. P values calculated using Fisher's exact test to compare transformation outcomes to LPKO + Vector. **** indicates a p value < 0.0001.

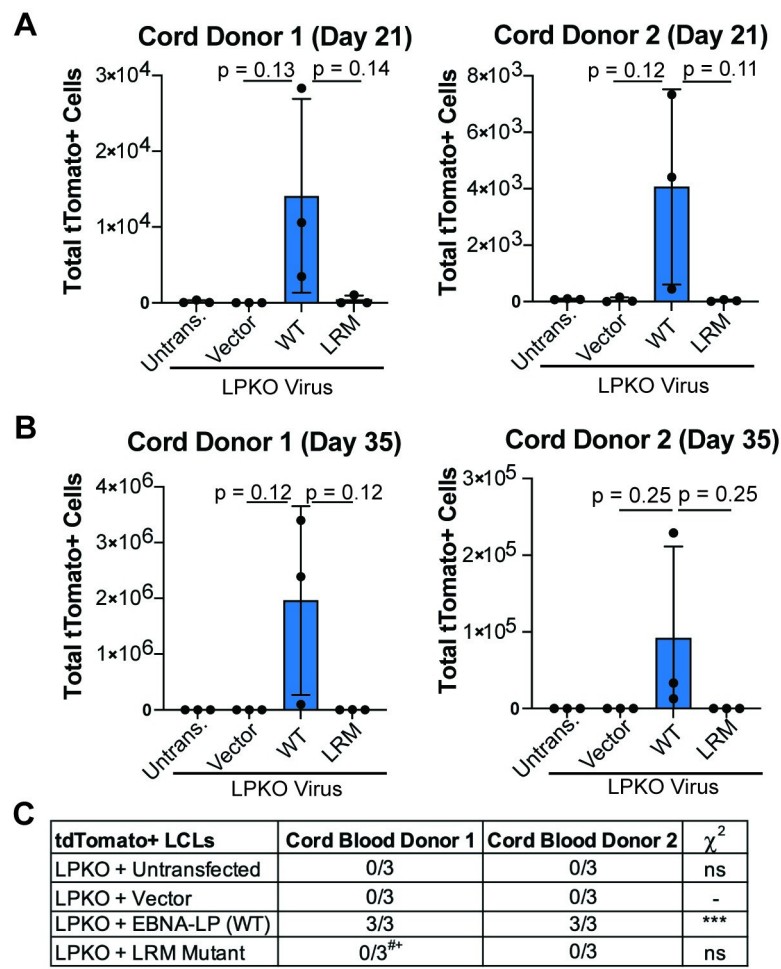

**Fig 4. EBNA-LP Leucine-rich motifs are important in transformation of cord blood B cells. A**. Total tdTomato positive cells at 21 days post infection for each cord blood donor in LPKO virus infected, trans-complemented cells (n = 3). P values from unpaired t-test. Mean and standard deviation are plotted. **B.** Total tdTomato positive cells from trans-complemented cord blood B cells at 28 or 35 days post infection weeks (n = 3). **C.** Number of tdTomato positive LCLs generated from trans-complemented LPKO infected cord blood B cells. #Indicates conditions in which some replicates generated tdTomato negative LCLs. +Indicates a tdTomato negative LCL which grew out slower compared to those trans-complemented with wild type EBNA-LP. P values calculated using Fisher's exact test to compare transformation outcomes to LPKO + Vector. *** indicates a p value < 0.001.

cells, LRM mutant transfected LPKO infections of total B cells yielded LCLs–presumably from transformation of memory B cells. Importantly, wild type and LRM mutant EBNA-LP were expressed at similar levels in these LCLs (Fig 5A). Furthermore, the LRM mutant was visible both in sub-nuclear puncta and diffuse within the nuclei of trans-complemented LPKO LCLs similar to wild type EBNA-LP (Fig 5B). These results suggest failure of the LRM mutant to contribute to naïve B cell transformation is not due to misfolding or failure to express protein.

## EBNA-LP and YY1 Extensively Associate on Chromatin in LCLs

Given the role of leucine-rich motifs in coactivation by PGC family members, we used publicly available ChIP-Seq data from LCLs to identify cellular transcription factors whose binding sites on the genome had significant overlap with those of EBNA-LP. We found that YY1, a transcription factor co-activated by PGC proteins, shared many sites with EBNA-LP in

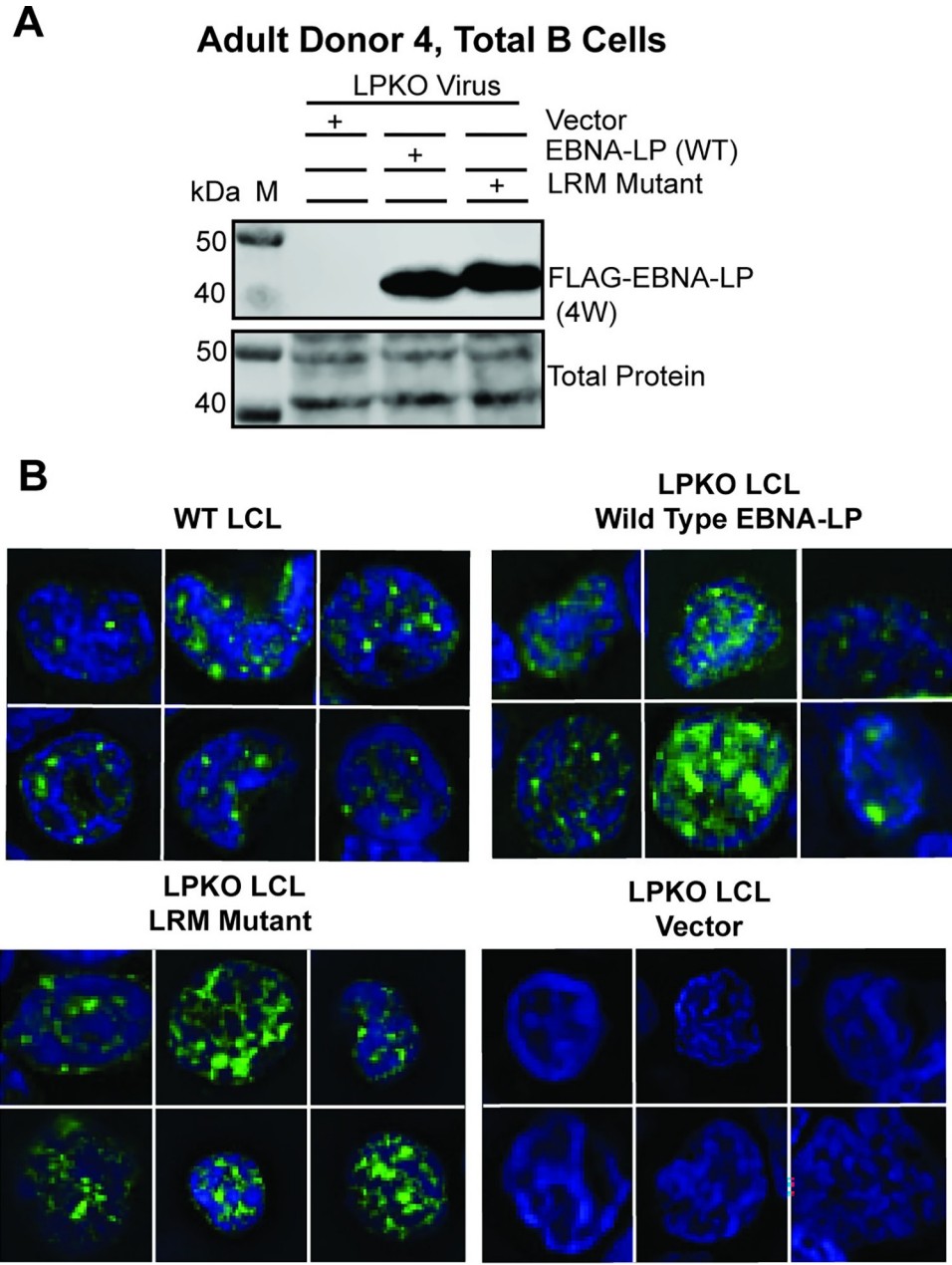

**Fig 5. EBNA-LP Leucine-rich mutant is expressed and localizes similar to wild-type EBNA-LP in LCLs derived from adult total B cell infection with LPKO virus. A.** LCLs from Adult Donor 4, total B cells infected with LPKO virus retain wild type and LRM mutant EBNA-LP expression. **B**. Immunofluorescence of total B cell derived LCLs, Adult Donor 5 WT LCL, and Adult Donor 4 LPKO derived LCLs trans-complemented with wild type EBNA-LP, LRM mutant EBNA-LP, or vector only. Green = Anti-EBNA-LP (JF186), Blue = DAPI.

agreement with previous work [18,44,45] (Fig 6A). EBNA2 also shares many overlapping sites with YY1 alone, and both EBNA-LP and YY1 (Fig 6A). While the sites shared by EBNA2, EBNA-LP, and YY1 occur primarily at intronic and intergenic regions (Fig 6B), the EBNA-LP and YY1 overlapping sites that lack EBNA2 binding occur to a higher degree at promoter regions (Fig 6C).

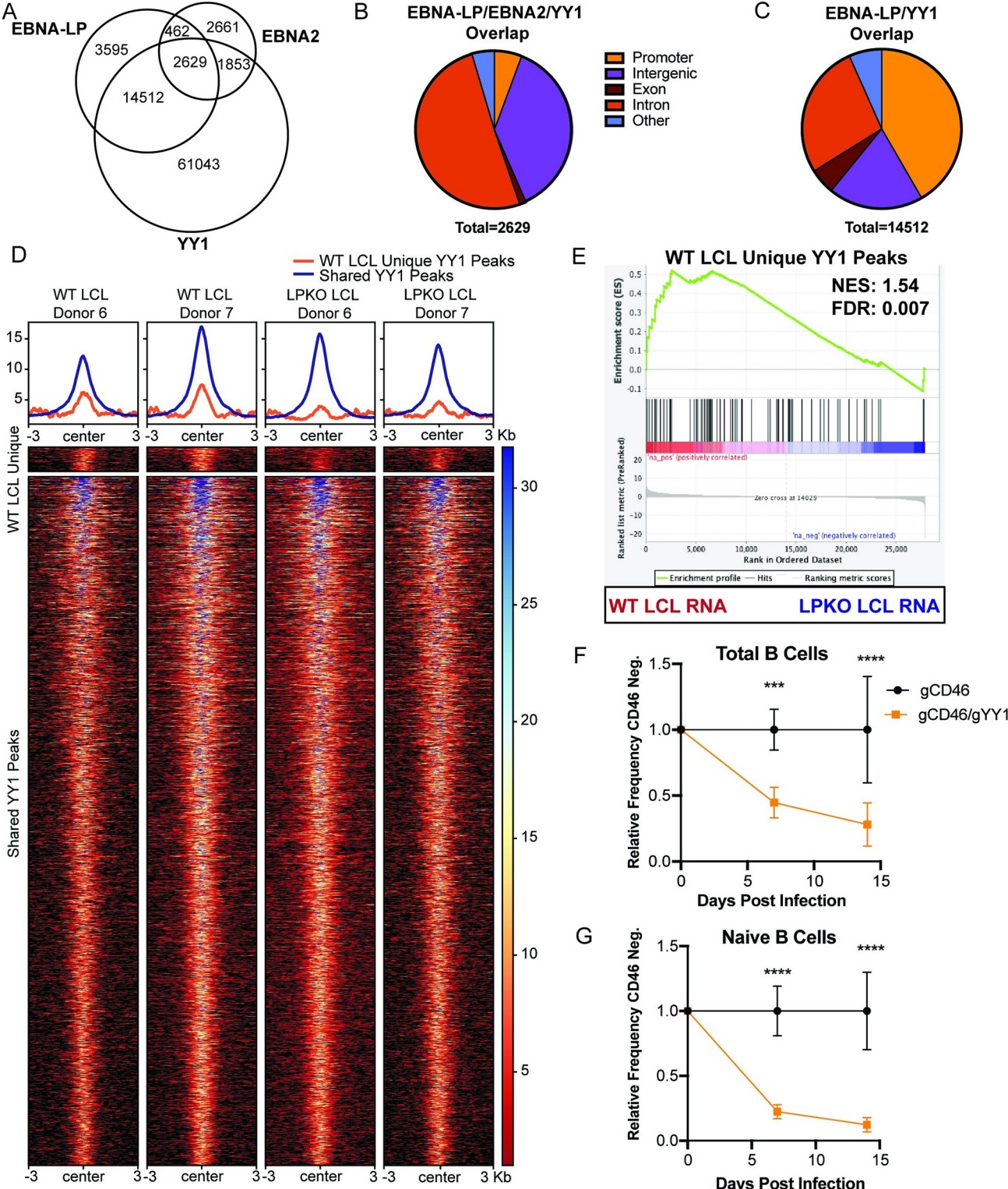

**Fig 6. EBNA-LP and cellular transcription factor YY1 association on chromatin in LCLs. A.** EBNA-LP and EBNA2 localized sites on chromatin overlap with YY1 in LCLs by publicly available ChIP-Seq data. **B.** Location on chromatin of overlapping peaks between YY1, EBNA-LP, and EBNA2. **C.** Location on chromatin of overlapping peaks between exclusively YY1 and EBNA-LP. **D.** Heatmap of YY1 peaks gained in WT LCLs compared to LPKO (dark blue), or shared across WT and LPKO LCLs (light blue) in Donors 6 and 7. **E.** Gene Set Enrichment Analysis of YY1 peaks unique to WT LCLs compared to RNA-seq gene expression of WT and LPKO LCLs. NES = normalized enrichment score. FDR = false discovery rate q-value. **F.** Relative

frequency of CD46 negative cells when total B cells were transfected with Cas9/gRNA complex targeting control surface marker CD46 or CD46 and YY1 prior to infection. **G.** Relative frequency of CD46 negative naïve B cells with CD46 or CD46 and YY1 knockout prior to infection. F and G experiments were statistically analyzed by two-way ANOVA followed by Sidak post-hoc test. *** indicates p-values < 0.001, **** indicates p-values < 0.0001.

We utilized two donor-matched WT and LPKO LCLs to determine whether EBNA-LP influences YY1 chromatin binding in LCLs. While the majority of YY1 peaks were shared across WT and LPKO LCLs, YY1 peaks unique to WT LCLs were identified (Fig 6D). Differential enrichment analysis further confirmed 83 of the WT LCL unique YY1 peaks were significantly enriched for YY1 binding in WT LCLs compared to LPKO LCLs. 78 YY1 peaks unique to LPKO LCLs were also identified. YY1 peaks were annotated by identifying the nearest transcription start site and assigning peaks to that gene using HOMER [46]. RNA-Sequencing data from three donor-matched pairs of WT and LPKO LCLs, including the same two donors used for YY1 Cut&Run, (S2 Table) was used to compare gene expression changes. The genes for which at least 1 unique YY1 peak was gained in WT LCLs were more likely to be highly expressed in WT LCLs compared to LPKO LCLs (Fig 6E), supporting a role for EBNA-LP in regulating YY1 chromatin-binding as a mechanism to activate gene expression.

A functional role for YY1 in EBV infection and B cell transformation has not previously been described. Therefore, we used a Cas9 RNP-based knock out approach to assess the importance of YY1 in EBV-infected B cell outgrowth. The non-essential cell surface protein CD46 was used as a control and proxy for successful gene targeting as previously described by others [47]. We found that YY1 knock out in either total B cells (Fig 6G) or naïve B cells (Fig 6H) significantly reduced EBV-induced proliferation early during infection compared to the CD46 knock out control and failed to generate YY1 knock out LCLs, however this effect was not EBV-specific as outgrowth of CpG-stimulated B cells was also reduced by YY1 loss (S7A Fig). These data support a functional role for EBNA-LP in modulating the association of YY1 with the genome at sites of YY1-regulated transcription during B cell transformation.

## Conserved Leucine-Rich Motifs in EBNA-LP are Required for Association with YY1

To determine whether YY1 and EBNA-LP interact, we performed endogenous YY1 co-immunoprecipitation in EBV B95-8 strain derived LCLs for endogenous EBNA-LP. In three LCLs, YY1 pulled down EBNA-LP, further supporting complex formation between YY1 and EBNA-LP (Fig 7A). As the interaction interface between EBNA-LP and YY1 is unknown, we investigated whether the leucine-rich motifs were indeed important. Using the trans-complemented LPKO LCLs derived from total B cells, reciprocal co-immunoprecipitations showed that wild type, but not LRM mutant EBNA-LP pulled down YY1 (Fig 7B), and conversely that YY1 co-immunoprecipitation pulled down wild type EBNA-LP, but not the LRM mutant protein (Fig 7C). Therefore, these LRM motifs in EBNA-LP that are required for transformation of naïve B cells are also required for interaction with the cellular transcription factor YY1, indicating a novel function for EBNA-LP in cellular transformation during EBV infection.

Using LPKO LCLs established from total B cells that had been transfected (prior to LPKO EBV infection) with wild type EBNA-LP, LRM mutant EBNA-LP, or the control vector, we examined YY1 binding on chromatin. Overall, LPKO LCLs trans-complemented with wild type EBNA-LP had more YY1 binding sites determined by calling peaks using SEACR (Fig 7D) [48]. The 671 core peaks that were common across all three conditions had an enrichment of YY1 binding in LCLs trans-complemented with wild type EBNA-LP compared to the LRM mutant or vector alone (Figs 7E and S8A). Genes close to YY1 peaks unique to the LPKO LCLs trans-complemented with wild type EBNA-LP compared to LRM mutant EBNA-LP are again

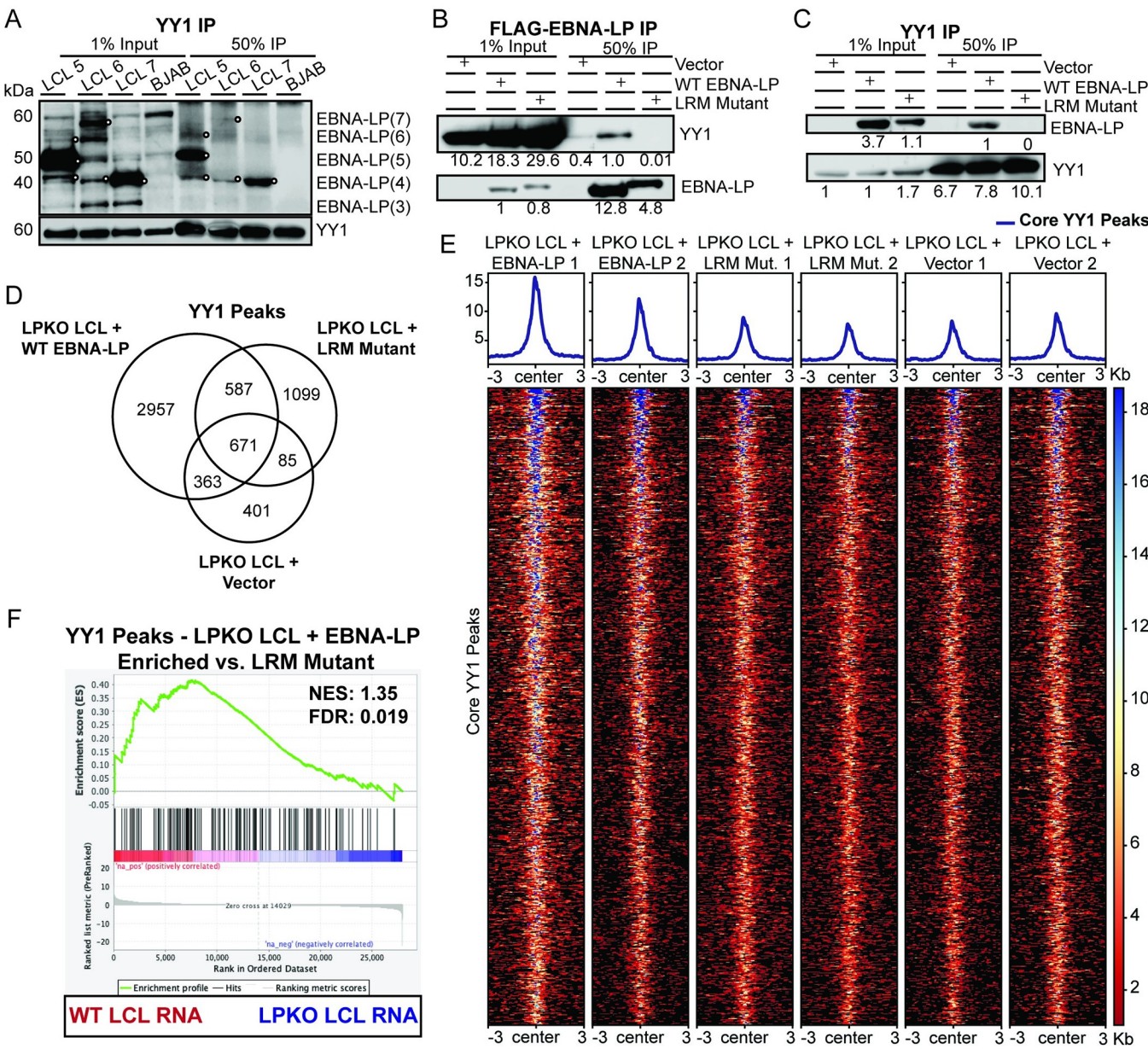

**Fig 7. EBNA-LP associates with YY1 cellular transcription factor through leucine-rich motifs. A.** Endogenous co-immunoprecipitation in LCLs to YY1. White circles indicate EBNA-LP specific bands. BJAB, an EBV-negative cell line, was used as a negative control. Signal in BJAB lanes indicate non-specific background bands. W indicates the number of W domain in the indicated isoform. **B.** FLAG-EBNALP co-immunoprecipitation in LPKO LCLs (Donor 4) generated from trans-complementation of total B cells with vector control, wild type, or LRM mutant EBNA-LP. Signal is quantified below band. **C.** YY1 co-immunoprecipitation in trans-complemented LPKO LCLs. **D.** Venn diagram of YY1 peaks identified by Cut&Run in LPKO trans-complemented LCLs (Donor 4). **E.** Heatmap of the identified 671 core YY1 binding sites shared across all three conditions. **F.** Gene set enrichment analysis of genes associated with YY1 peaks that are significantly enriched in LPKO LCLs trans-complemented with wild type EBNA-LP compared to LRM mutant against the differentially expressed genes in WT and LPKO LCLs. NES = normalized enrichment score. FDR = false discovery rate q-value.

more likely to be upregulated in RNA-Sequencing data for WT LCLs compared to LPKO LCLs unlike the LRM mutant unique peaks (Figs 7F and S8B–S8C). However, these significantly enriched YY1 peaks were not entirely identical to those we identified in WT LCLs. Unique peaks were also identified in LRM mutant and vector trans-complemented LCLs, but these did not significantly correlate with WT or LPKO LCLs enriched genes (S8D–S8I Fig).

These findings support our biochemical and mutagenesis data, suggesting EBNA-LP modulates YY1 chromatin-association in an LRM-dependent manner.

## Discussion

EBV viral proteins aid in maintaining life-long, latently infected B cells by facilitating the proliferation and survival of infected cells. When the virus-host balance is disturbed, such as in the absence of a functional immune system, the characteristics of these viral proteins can contribute to oncogenesis as modeled by *in vitro* transformation of primary B cells by EBV. While EBNA-LP is essential for the transformation of naïve B cells by EBV, it has primarily been studied outside of the context of infection. As EBNA-LP is encoded in the viral genome across multiple repeat units of the major internal repeat, making mutations to the EBNA-LP coding sequence in the viral genome is particularly challenging and time-consuming. Thus, we have developed a trans-complementation system by which the well-conserved regions of EBNA-LP could be assessed in the context of primary B cell infection.

Excitingly, we could reliably transfect resting primary B cells with an episomal construct that was maintained upon EBV infection and outgrowth into LCLs. This assay could therefore be used to screen a large number of EBNA-LP mutants in the context of EBV infection of primary B cells, beyond those we present here. This trans-complementation strategy presents a system by which it is possible to screen important functions of EBNA-LP so that the more precise but technically challenging and time-consuming approach of constructing mutant viruses can be more intelligently targeted. It also opens up the possibility of trans-complementing with proteins from other viruses that target the same biological processes, and can be adapted for trans-complementing EBV mutants more generally. Therefore, by pairing infection of knock out viruses with trans-complementation of mutant viral proteins, this assay could be applied to identify key features of other viral proteins in EBV transformation of primary B cells.

Unexpectedly–given the importance of EBNA-LP phosphorylation in EBNA2 co-activation assays–we found that EBNA-LP phosphorylation does not contribute detectably to the outgrowth of LCLs, similar to previous work showing no evidence of LPKO infected B cells having reduced induction of cellular genes by EBNA2 [32]. As such, our work highlights both the limitations of studying viral proteins outside of the context of primary infection and suggests EBNA-LP has numerous roles in the survival and transformation of infected naïve B cells. However, *in vitro* primary B cell infection may not fully recapitulate *in vivo* EBV infection as high ratios of virus to cells, in our system 0.2 Raji Green Units was equivalent to approximately 50–100 encapsidated viral genomes per cell, are required for infection of almost all cells. As such, high gene dosage of viral proteins may partially mask EBNA-LP activities *in vitro*. However, while EBNA-LP phospho-mutants trans-complemented LPKO virus as effectively as wild type EBNA-LP, we instead identified highly conserved leucine-rich motifs of the sequence LXXLL or LXXXLL in both the W domain CR1 and Y domain CR4 that are required to rescue LPKO infected naïve B cells derived from adult peripheral blood and cord blood While the presence of small numbers of antigen-experienced B cells–both conventional and double negative (IgD-/CD27-) memory B cells–in our adult peripheral blood naïve B cell fractions allowed the outgrowth of some tdTomato negative LCLs, our results indicate that the leucine-rich motifs in EBNA-LP play an important role in EBV-mediated transformation. LXXLL and similar motifs have been found in the effector domains of numerous viral proteins involved in transcriptional regulation [49], although a role for this motif in EBNA-LP has not been explored. We found the conserved leucine-rich motifs in EBNA-LP mediate interaction with YY1. YY1 is both a transcription factor and DNA looping factor that promotes interactions

between enhancer and promoter regions [50]. YY1 is important in B cells as a transcriptional regulator of the germinal center reaction [51]. In the absence of YY1, naïve B cells are unable to differentiate into germinal center B cells or plasma cells [52–54]. The requirement for YY1 may in part be due to regulation of apoptosis, as YY1 knockout in pro-B cells and germinal center B cells causes these cells to be more prone to apoptosis [52,54]. YY1 deficient naïve B cells also display reduced proliferation upon stimulation compared to controls [52]. YY1 has also been implicated in transcriptional regulation of genes involved in oxidation phosphorylation in B cells [52,53,55], upregulation of which is required to avoid cellular arrest upon EBV infection [8]. As EBV-infected cells undergo B cell activation and differentiation in a manner that mimics the remodeling of antigen-experienced B cells through the germinal center [56,57], YY1 may therefore be required for survival and effective cellular remodeling during EBV infection and transformation.

We performed YY1 Cut&Run in WT, LPKO, and LPKO trans-complemented LCLs to examine the role of EBNA-LP in regulating YY1 chromatin-association. LCLs were used given the technical limitations of performing this assay early during infection, in particular the low viability of LPKO infected B cells early during infection compared to WT and the low transfection efficiency of primary B cells. However, LPKO LCLs derived from Pan B cells come from memory B cells that in some way compensate for the absence of EBNA-LP unlike naïve B cells, and as such observed differences may in part be attributed to distinct functions of YY1 in the survival of naïve but not memory B cells–and more distinct differences in EBNA-LP-dependent YY1 activity may occur earlier during EBV infection of naïve B cells. Further investigation of YY1 activity in naïve and memory B cells and early during EBV infection will allow for a more complete understanding of the importance of EBNA-LP in modulating YY1 function in naïve B cells.

Despite these limitations, distinct YY1-associated regions were detected in WT LCLs compared to LPKO LCLs at sites correlated with more active gene transcription in WT LCLs. In LPKO LCLs trans-complemented with wild type EBNA-LP compared to the LRM mutant or vector we observed a general increase in YY1 binding sites and overall strength of YY1 chromatin-binding. We again detected YY1 peaks significantly enriched in LPKO LCLs trans-complemented with wild type EBNA-LP compared to mutant at sites of more activated gene expression in WT LCLs compared to LPKO LCLs. However, as these genes identified in the trans-complementation system are not identical to those in WT LCLs, we do not have clear evidence in this trans-complementation system that EBNA-LP is consistently required for YY1 association at a specific set of genes. This may in part be due to the trans-complementation system having more background than with wild type and mutant virus transformed LCLs, as despite best efforts to sort for tdTomato positive cells, the trans-complementation plasmid is lost over time in LCLs with less selective pressure for EBNA-LP than early during infection, and thus it is possible not all trans-complemented cells express EBNA-LP unlike the WT virus transformed LCLs. Instead, we propose that EBNA-LP may play a more modulative role in recruiting or stabilizing YY1 on chromatin in order to activate gene transcription.

Furthermore, the unique importance for EBNA-LP in naïve and not memory B cell transformation upon EBV infection, suggests basal differences between naïve and memory B cells exist that require different strategies for viral transformation. Naïve B cells, for example, may require additional viral assistance in avoiding cell death compared to memory B cells, as naïve B cells express lower levels of antiapoptotic proteins and are more likely to undergo apoptosis in both the presence and absence of antigen stimulation than memory B cells [58]. Memory B cells also proliferate more rapidly upon antigen activation than naïve B cells [59,60] as naïve B cells express elevated levels of negative cell cycle regulators [59]. Memory cells are also more predominantly in the cell cycle phase $G_1$ compared to naïve B cells [61], suggesting memory B

cells are better primed to enter cell division while naïve B cells may have greater intrinsic barriers to EBV induction of proliferation. Further, activation of naïve B cells by antigen increases both glycolysis and oxidative phosphorylation [62] and requires mitochondrial remodeling [63]. While little is known about the metabolic differences between resting naïve and memory B cells, memory T cells have increased mitochondrial mass compared to naïve T cells, and also have a greater capacity for both oxidative phosphorylation and glycolysis upon activation suggesting the transition from naïve to memory in immune cells alters metabolic capacity long term [64]. Therefore, EBNA-LP association with YY1 may be important in naïve B cells to overcome barriers to transformation including apoptosis, initiation of proliferation, and cellular metabolism which may be innately more restrictive in naïve B cells than memory B cells. B cell activation and differentiation also requires a large degree of chromatin remodeling, to which YY1, as a chromatin looping factor, may contribute. During the transition from naïve B cells to germinal center B cells, roughly 95% of chromatin compartments are remodeled, primarily to a more active chromatin state, followed by 73% of remodeled regions returning to their original states in memory B cells [65]. EBV infection largely mimics this chromatin remodeling process [66]. Still, distinctions between the chromatin architecture of naïve and memory B cells have not been well characterized. For example, whether memory B cells have higher accessibility of YY1 motifs compared to naïve B cells remains to be studied, but could provide insight as to further differences in naïve B cell dependence on EBNA-LP and YY1 association.

As YY1 may be important in modulating gene expression of EBV-infected cells, the virus may have multiple strategies to manipulate YY1 chromatin association. As a mechanism for cellular gene regulation, YY1 has been implicated in the formation of "EBV super-enhancers" on host chromatin, which are defined as regions of accessible chromatin where multiple EBNAs and cellular transcription factors associate leading to increased expression of target genes [67]. "EBV super-enhancers" target genes important in infected cell survival and proliferation including Myc [67]. YY1 has is also important in inducing Bcl2a1 (BFL-1) expression in LCLs, promoting resistance to apoptosis [68] and we also observe a high degree of EBNA2 overlap with YY1 in agreement with previous work [18,69]. Therefore, EBNA-LP association with YY1 for genome rearrangement may in fact be a mechanism by which EBNA-LP stabilizes induction of important cellular genes and viral genes along with other viral transcriptional regulators.

YY1 is also able to alter the structure and epigenetic marks of numerous DNA virus genomes resulting in both activation and repression of viral genes [70]. These viruses include herpesviruses such as Human Cytomegalovirus and Kaposi's Sarcoma-Associated Herpesvirus, small DNA tumor viruses including Human Papilloma Virus, and many others [70–74]. In the EBV genome, YY1 binds and represses the promoters of the immediate early genes BZLF1 and BRLF1, preventing induction of the productive replication cycle [75–77]. There is also a YY1 binding site upstream of the immediate-early latency promoter Wp [78]. This region including the YY1 binding site is important for transcription from both the Cp and initial Wp, and deletion hinders transformation of infected cells [79]. Therefore, YY1 binding to the viral genome may contribute to inducing adequate expression of viral latency proteins through regulation of viral Wp and Cp promoter usage during early infection and transformation. Whether YY1 occurrence at these cellular and viral genomic sites is EBNA-LP dependent requires further investigation.

Additionally, motifs similar to the leucine rich motifs we identified in EBNA-LP, including LXXLL motifs, have been found in numerous families of viral proteins involved in transcriptional regulation [49]. While we have demonstrated that the LRM mutant disrupts YY1 interaction, whose impact can be seen in the changes in YY1 recruitment and gene activation, we

cannot exclude that interactions with other cellular proteins might also be disrupted by this mutant. For example, EBNA-LP is known to engage and re-localize the PML-body component SP100 [20,80] and associate with cellular proteins including DNA-PK, AKAP8L/HA95 [17,28], and numerous pre-mRNA processing factors among others [81,82]. As such, whether EBNA-LP engages these additional transcriptional regulators through this repeated leucine-rich motif may provide further insight as to the mechanism by which EBNA-LP regulates gene expression of EBV-infected cells.

Overall, this study identifies previously uncharacterized, yet highly conserved, motifs within EBNA-LP that are important for transformation of EBV-infected naïve B cells. This work suggests EBNA-LP encodes leucine-rich motifs to engage the important cellular transcription factor YY1 for EBV-induced cellular transformation.

## Materials and methods

### Virus preparation

LPKO virus refers to LPKO^w from Szymula et al. [32] encoding stop codons in exon W2 of every internal W repeat. Wild Type EBV (WT) refers to the WT^w from Szymula et al. [32] that has a repaired stop codon in one repeat of EBNA-LP that is found in the parental B95-8 genome [83], and encodes a total of 6 W repeats. Viruses were prepared from 293-EBV producer lines and Raji Green Unit titer was obtained as previously described [32], using flow cytometry to count green cells [32,84]. B95-8 strain of EBV was produced from the B95-8 Z-HT cell line as previously described [85].

### Cell culture and cell isolation

Adult buffy coats were obtained from the Gulf Coast Regional Blood Center (Pro00006262) and cord blood was obtained from the Carolinas Cord Blood Bank (Pro00061264). Peripheral blood mononuclear cells were isolated using Histopaque-1077 (Sigma, H8889) and SepMate PBMC Isolation tubes. Naïve B cells were then isolated from adult blood using the EasySep Human Memory B Cell Isolation kit (STEMCELL Technologies #17864) to first remove CD27 positive memory cells, followed by negative isolation of B cells. To obtain total B cells, the Easy Human Pan-B Cell Enrichment kit (STEMCELL Technologies #19554) was used. Purity of isolated cells was assessed by flow cytometry with antibodies to CD19, IgD, and CD27 (Table 1). Primary cells were maintained in Roswell Park Memorial Institute (RPMI) medium 1640 supplemented with 20% heat-inactivated fetal bovine serum (FBS) (Corning). LCLs and BJAB cell lines were maintained in RPMI supplemented with 10% FBS and 2 mM L-glutamine, penicillin/streptomycin. 293T cells were maintained in Dulbecco's Modified Eagle Medium (DMEM) with 10% FBS.

### DNA constructs

EBNA-LP cDNA was synthesized (GENEWIZ) to encode wild type (B95-8) EBNA-LP or mutants containing four copies of the W repeat domain (S1 Table). Codon usage of this repeat was modified such that the amino acid sequence in each W domain was identical, but the DNA sequence in each W domain was unique in order to avoid recombination of repeated W domains and facilitate downstream cloning. S3A and S3E EBNA-LP genes were synthesized with the S34, S36, and S63 in each W domain modified to alanine or glutamic acid respectively. For the LRM mutant, EBNA-LP cDNA was first synthesized such that L25, L28, and L29 in each W domain were modified to alanine. QuikChange XL Site-Directed Mutagenesis (Agilent, 200517) was then used to modify L13, L17, and L18 in the Y domain to alanine.

**Table 1. Reagents and materials used in methods.**

| Name | Company | Catalog. No. | Concentration |
|---|---|---|---|
| JF186 Antibody (EBNA-LP) | Purified in house to 1 mg/mL | NA | Western blot 1:500, Immunofluorescence 1:100 |
| YY1 Monoclonal Antibody | Proteintech | 66281-1-Ig | 1:500, IP |
| YY1 (D5D9Z) | Cell Signaling Technology | 46395S | 1:1000, western blot, 1 µL Cut&Run |
| YY1 antibody (pAb) | Active Motif | 61779 | 1:1000, western blot |
| Anti-Mouse IgG-Peroxidase | Sigma-Aldrich | A9044-2ML | 1:5000, western blot |
| Anti-Rabbit IgG-Peroxidase | Sigma-Aldrich | A0545-1ML | 1:3000, western blot |
| VeriBlot | Abcam | ab131366 | 1:500, western blot |
| Alexa Fluor 488 goat anti-mouse IgG (H+L) | ThermoFisher Scientific | A11001 | 1:250, immunofluorescence |
| Normal Mouse IgG1 | Santa Cruz | sc-3877 | 5 ng per IP |
| Normal Mouse IgG2a | Santa Cruz | sc-3878 | 5 ng per IP |
| Anti-FLAG M2 Antibody | Sigma-Aldrich | F1804 | 1:100, IP |
| CD46 PE | BioLegend | 352402 | 1:100, Flow Cytometry |
| IgD PE/Cy7 | BioLegend | 348217 | 1:100, Flow Cytometry |
| CD27 AlexaFluor 488 | BioLegend | 393204 | 1:100, Flow Cytometry |
| CD19 APC | BioLegend | 302212 | 1:100, Flow Cytometry |
| cOmplete, mini, EDTA-free Protease Inhibitor Cocktails | Roche | 4693159001 | 1x protein lysates, 2x IP |
| PhosSTOP | Roche | 4906837001 | 1x |
| CD46 Guide #1 | SYNTHEGO | GAGAAACAUGUCCAUAUAUA | |
| CD46 Guide #2 | SYNTHEGO | AACUCGUAAGUCCCAUUUGC | |
| CD46 Guide #3 | SYNTHEGO | UUGCUCCUUAGAGGAAAUAA | |
| YY1 Guide #1 | SYNTHEGO | CUGGUCACCGUGGCGGCGGC | |
| YY1 Guide #2 | SYNTHEGO | GGAGGCGGCCGCGUCAAGAA | |
| YY1 Guide #3 | SYNTHEGO | CGACCCGGGCAACAAGAAGU | |

EBNA-LP constructs were first cloned into pSG5-FLAG-Gateway (a gift from Eric Johannsen) for expression in 293T cells by amplifying with primers encoding attB1/attB2 recombination sites and performing recombination by gateway cloning (ThermoFisher 11789020, and 11791020). For trans-complementation assays, FLAG-tdTomato-P2A was cloned into pCEP4 vector by restriction digest cloning. FLAG-EBNA-LP constructs were then cloned into this vector using XhoI and BamHI restriction sites immediately following the P2A cleavage site (S1 Fig). Following P2A cleavage, EBNA-LP protein includes a proline (site of cleavage), FLAG epitope, linker including the attB1 recombination site from gateway cloning, and full-length EBNA-LP (S1 Fig).

## Trans-complementation assays

Purified adult naïve or cord blood B cells were resuspended in Belzer UW Cold Storage Solution (BridgetoLife) at 480,000 cells/7 µL and combined with 1.2 µg of DNA in 5 µL Buffer T (Neon Transfection Kit) for excess volume to avoid air bubbles. As 10 µL tips were used for transfection, approximately 400,000 cells total were transfected per replicate. DNA was prepared by maxi prep (ZymoPURE, D4203), eluting DNA in Buffer T (Neon Transfection Kit). Cells were transfected using the Thermo Fisher Neon Transfection System at 2150 V, 20 ms, 1 pulse and resuspended in 200 µL RPMI with 20% FBS. One hour after transfection, cells were infected with LPKO or WT virus at 0.2 Raji Green Units per cell for 1 hour at 37°C, a ratio we previously determined by EBNA2 immunofluorescence to infect almost all B cells and an optimal ratio for outgrowth, similar to previous reports [9]. As quantified by BALF5 DNA qPCR, this ratio of RGU is equivalent to approximately 50 LPKO and 100 WT virus genomes per cell.

Every seven days, cell outgrowth was quantified by flow cytometry. Media was added during outgrowth upon increased cell density.

## Flow cytometry

Flow cytometry was used to assess purity of isolated primary cells and to quantify outgrowth of trans-complemented naïve B cells. For sample preparation following isolation, cells were washed once in FACS Buffer (PBS with 2% FBS), stained with indicated antibodies (Table 1) for 15 minutes in the dark at room temperature, then rinsed again with FACS buffer. For samples in the trans-complement assays, samples were prepared and collected in a 96-well V-bottom plate. First, samples were evenly resuspended, and a known volume of each sample was moved to the plate. Samples were then washed once in FACS buffer and resuspended in 100 μL of FACS buffer prior to collection. Samples were collected using the BD FACSCanto-II with a high throughput sampler to collect precise sample volumes. Analysis was performed using FlowJo software (FlowJo, LLC).

## Amino acid sequence alignment

Alignment was performed with Geneious Prime 2023.1.2 (https://www.geneious.com) multiple sequence alignment. The LCV Baboon, Gorilla, and Chimpanzee strains were previously published [24]. The remaining sequences were obtained from NCBI (NC_007605.1, NC_006146.1, NC_009334.1). Amino acid number was specified from internal W domain amino acid sequence for consistency, rather than numbered from the start codon.

## 293T FLAG immunoprecipitation

293T cells were transfected with 20 μg DNA followed by media change the next day. Three days post-transfection, cells were washed twice with cold PBS followed by addition of lysis buffer (150 mM NaCl, 0.5% Sodium Deoxycholate, 1% NP-40, 50 mM Tris pH 8, 10% glycerol, 2x protease inhibitor, 1x phosphatase inhibitor (Table 1). Cells were scraped and lysates were shaken for 30 minutes at 600 rpm at 4°C followed by centrifugation at maximum speed for 10 minutes. Lysates were pre-cleared with Dynabeads Protein G (ThermoFisher, 10004D) conjugated to Normal Mouse IgG (Table 1). Lysates were quantified by Bradford assay (Bio-Rad, 5000006) and 1 mg of protein was used per condition. Lysates were rotated with antibody overnight with rotation. Dynabeads were added and incubated for 3 hours at 4°C while rotating. After incubation, beads were washed four times in wash buffer (150 mM NaCl, 0.5% NP-40, 50 mM Tris pH 8, 10% glycerol, 1x protease inhibitor, 1x phosphatase inhibitor) and eluted in 1x LDS Sample Buffer (ThermoFisher, NP0007) by boiling 10 minutes at 55°C.

## Co-Immunoprecipitations in LCLs

40–50 million cells were pelleted, washed once in PBS, and resuspended in cold lysis buffer (150 mM NaCl, 50 mM Tris pH 7.6, 1% NP-40, 2 mM EDTA, sodium molybdate, 2x protease inhibitor, 1x phosphatase inhibitor). Lysates were prepared as described above for FLAG Immunoprecipitations. Input was stored with 1x LDS Sample Buffer after pre-clear. Beads were washed four times with fresh, ice-cold lysis buffer (YY1 co-immunoprecipitations in B95-8 derived LCLs) or lysis buffer with 0.5% NP-40 (FLAG and YY1 co-immunoprecipitations in LPKO trans-complemented LCLs). Samples were eluted in 1X LDS Sample Buffer by boiling at 37°C for 10 minutes.

## Western blotting

Cells were washed one time with PBS before pelleting. Pellets were resuspended in 1x RIPA buffer with protease and phosphatase inhibitors added. After shaking samples at 4˚C at 600 rpm, samples were centrifuged for 10 minutes at 14,000xg. The lysates were then quantified using Bradford assay and diluted to 1x LDS. Samples were loaded on a 4–12% Bis-Tris gel (ThermoFisher, NP0321BOX) and run in MOPS Running Buffer (ThermoFisher, NP0001) followed by transfer to PVDF membranes using a TransBlot Turbo Transfer System at 25 V, 1.3 Å, for 12 min (Bio-Rad). Total protein was quantified using total protein stain (LI-COR, 926–11011) and blocked in 5% milk in 1x TBST. Blot was incubated overnight with primary antibody in 1x TBST with 5% BSA at 4˚C. Blots were then washed in 1x TBST, incubated in secondary antibody for 1 hour at room temperature in 5% milk, and washed three times for 10 minutes. Blots were incubated with HRP substrate and imaged on the LI-COR Odyssey XF Imager.

## Immunofluorescence

Immunofluorescence was performed as described previously [8]. Slides were mounted in VECTASHIELD HardSet Antifade Mounting medium with DAPI (H-1500-10). Slides were imaged using the Olympus IX81 microscope using a 60X oil objective.

## ChIP-Seq analysis

YY1 ChIP-Seq data aligned to hg19 was obtained from ENCODE (Experiment ENCSR000BNP) and EBNA-LP ChIP-Seq data was a kind gift from Dr. Bo Zhao [18]. Bam files were used to call peaks in MACS2 using corresponding IgG ChIP-Seq data as background. EBNA2 ChIP-Seq peaks called by MACS2 aligned to hg19 from two replicates were obtained from NCBI GEO (GSM5360537, GSM5360538) [69]. HOMER [46] was used to identify EBNA2 peaks conserved between replicates which were then used for downstream analysis. In HOMER, the function mergePeaks with the default distance -d given was used to identify over-lapping and uniquely bound sites between YY1 and EBNA-LP. The function annotatePeaks.pl was then used to assign overlapping peaks to genomic regions.

## Cut&Run

Cut&Run was performed using the CUTANA ChIC/CUT&RUN kit (Epicypher 14–1048) with the following modifications. For LPKO trans-complemented LCLs, cells were sorted to 100% tdToamto positivity using a Sony SH800 and then expanded to reach required number of cells. All samples were performed with at least 2 biological replicates. 500,000 cells were used for each sample and cells were moderately cross-linked in 1% formaldehyde for 1 minute upon collection. Pre-Wash buffer was supplemented with 1% Triton X-100 and 0.05% SDS. Briefly, fixed cells were bound to ConA beads and incubated with YY1 antibody (46395S) or control IgG (provided in kit) overnight on a nutator at 4˚C. pAG-MNase was then bound to antibodies and chromatin digestion was performed for 2 hours at 4˚C. After collecting released DNA, cross-links were reversed by Proteinase K incubation overnight at 55˚C. DNA was puri-fied using provided columns. Libraries were prepared using the CUTANA CUT&RUN Library prep kit (Epicyhper 14–1001) according to protocol instructions with modifications. End repeat was modified to 20˚C for 20 minutes followed by 50˚C for 1 hour. PCR clean up beads (CNGS-0001) were used at 1.25x after ligation and 1.1x after indexing PCR to enrich for smaller fragments. Libraries were sequenced using a NextSeq 1000 P1 with 50 bp paired end reads.

## Cut&Run analysis

After sequencing, reads were aligned to hg38 with the concatenated EBV genome and the spike-in reference genome (E. coli K12) using the nf-core/cutandrun pipeline [86]. Bam files were then sorted using samtools and prepared for SEACR by converting to bed and bedgraph files. Bedgraph files were normalized by total read depth. Peaks were called using SEACR using the setting "norm" and "relaxed" [48]. Cut&Run samples performed with control IgG were used for background. HOMER was used to compare overlap of SEACR-called peaks using the mergePeaks function with default parameters "d -given" to look for literal overlap.

Heat plots were generated using deepTools computeMatrix and plotHeatmap function [87]. To generate heat plots for YY1 peaks in WT LCLs vs LPKO LCLs, separate bed files containing peaks that were shared in all four samples (Shared YY1 peaks) or shared by only WT LCL replicates (WT LCL Unique) were used to define reference point. Normalized bigwig files from each sample were then used as input. For the LPKO trans-complemented LCLs, the called peaks from two replicates of each condition were concatenated and the 671 identified overlapping peaks were used as reference points when generating the matrix. Additional replicates for LPKO LCLs trans-complemented with wild type EBNA-LP and control vector as shown in S8 Fig. Normalized bigwig files from each sample were used as input for read depth.

Peaks were annotated for the nearest gene using HOMER annotatePeaks.pl with the hg38 reference genome. The merged peaks for all samples were then concatenated into a single SAF file, which was used for differential gene expression by featureCounts [88] and DESeq2 [89]. Identified differentially bound genes were used to create a gene list for Gene Set Enrichment Analysis (GSEA) [90]. Significantly enriched peaks were defined as a log2fold change of greater than 0.7 and p-value less than 0.05.

## RNA-Sequencing

Cells were collected and dead cells were removed using a Ficoll gradient. RNA was isolated using the mini RNeasy kit (Qiagen 74104) including on-column DNase digestion. PolyA mRNA (E7490S) was isolated and libraries were made with the NEBNext Ultra II RNA Library Prep kit (E770S) and indexed (E7500S). Libraries were sequenced on a NovaSeq X Plus for 50 bp paired end reads. Reads were aligned using Hisat2 with the Hisat Grch38 (hg38) index. Samtools was then used to generate bam files which were used in DESeq2 for differential gene expression analysis to generate a ranked list of genes for GSEA.

## Cas9-RNP transfection

TrueCut Cas9 protein-v2 (ThermoFisher, A36499), 10 pmol per target, was incubated with 30 pmol sgRNA (Gene Knockout Kit v2, Synthego) per target (Table 1). Isolated naïve and total B cells were washed in PBS and resuspended in Buffer T (Neon Transfection System). 350,000 cells were transfected with Cas9/RNP complexes and infected with B95-8 virus as described above. Outgrowth was assessed by flow cytometry upon staining cells with anti-CD46 (Table 1).

### Mass spectrometry

See S1 Text for extended methods.

## Supporting information

**S1 Text. Extended methods for mass spectrometry identification of EBNA-LP phosphorylation sites and quantification.**
(DOCX)

**S1 Table. DNA Sequences of EBNA-LP constructs inserted into trans-complementation vector.**
(DOCX)

**S2 Table. RNA-sequencing expression data set of WT and LPKO LCLs.**
(CSV)

**S1 Fig. Construction of complementation vector. A.** Plasmid map of pCEP4 vector encoding FLAG-tdTomato, followed by a P2A cleavage site, and FLAG-EBNA-LP. **B.** Amino acid sequence of EBNA-LP expressed upon P2A cleavage. FLAG tag and residues encoding EBNA-LP are underlined.
(TIF)

**S2 Fig. Flow cytometry was used to assess tdTomato positive cells in trans-complementation assay. A.** Purity of CD19+ B cells in the naïve B cell population from adult blood donors after immunomagnetic selection prior to infection. Naïve B cells are defined as IgD+/CD27-. **B.** Gating strategy for identifying tdTomato+ cells at each time point. The virus encodes GFP, so this gating strategy is used to remove the majority of false tdTomato+ positives cells from GFP expressing cells. Representative samples from Donor 2 at 35 days post infection are shown. **C.** Total cells in each condition 14 days post infection for each donor (n = 3). Mean and standard deviation are plotted. P values are determined by unpaired t-test. **D.** Total cells in each condition at 28 or 35 days post infection (n = 3). **E.** In Donor 1 and Donor 3, tdTomato negative LCLs grow out in some replicates of LPKO infected untransfected cells, and LPKO infected cells transfected with control vector. Samples from Donor 3 at 35 days post infection are shown. Untransfected conditions were used to define threshold of background for tdTomato negative LCLs. Note some conditions did not lead to outgrowth of LCLs. *Indicates conditions in which cells were transformed into tdTomato negative LCLs.
(TIF)

**S3 Fig. Outgrowth of LPKO infected memory B cells is not impacted by trans-complementation with wild type EBNA-LP. A.** Purified of isolated memory B cells. **B**. Total tdTomato positive cells 14 days post infection. **C.** Total tdTomato positive cells 35 days post infection. Significance determined by unpaired t-test.
(TIF)

**S4 Fig. S34, S36, and S63 are all phosphorylated in EBNA-LP when expressed in 293T cells.** Tandem mass spectra from high-energy collisional dissocation (HCD) fragmentation localizing phosphorylation at sites S34 (**A**), S36 (**B**) and S63 (**C**) collected on a Fusion Lumos Orbitrap mass spectrometer. Localization using site-localizing y/b fragment ions were confirmed within Scaffold PTM using the AScore localization algorithm.
(TIF)

**S5 Fig. Trans-complementation of the LPKO virus with the LRM Mutant EBNA-LP fails to generate tdTomato positive LCLs. A**. Total cells in each condition 14 days post infection for each donor (n = 3). Mean and standard deviation are plotted. P values are determined by unpaired t-test. **B.** Total cells in each condition at 28 or 35 days post infection (n = 3). **C.** LCLs at 35 days post infection from adult blood donors. Single cell populations are plotted for tdTomato expression at 4–5 weeks post infection. Percentage of cells that are tdTomato positive as indicated. Note that some conditions did not lead to outgrowth of LCLs. *Indicates tdTomato negative LCLs.
(TIF)

**S6 Fig. Trans-complementation of LPKO infected cord blood with the EBNA-LP LRM Mutant fails to generate tdTomato positive LCLs. A.** Total cells in Cord Blood Donors in each condition 21 days post infection. Mean and standard deviation are plotted. P values determined from unpaired t-test. **B**. Total cells in Cord Blood Donors 35 days post infection. **C**. LCLs at 35 days post infection from Cord Blood Donors 1 and 2, at 5 weeks post infection. Percent indicates percent tdTomato positive cells. Note that no cells were transformed into LCLs in either donor from LPKO infected conditions that were either untransfected or transfected with control vector. *Indicates tdTomato negative LCL. **D.** Western blot of LCLs derived from cord blood donor 1.
(TIF)

**S7 Fig. CpG-stimulated B cell outgrowth is impacted by YY1 knockout. A.** Relative number of CD46 negative cells upon EBV infection or CpG stimulation 7 days post infection with CD46 alone or CD46 and YY1 knockout. P-values were calculated using Tukey's multiple comparisons test. ** Indicates p-values <0.01.
(TIF)

**S8 Fig. YY1 binds sites significantly enriched in LPKO LCLs trans-complemented with LRM Mutant or control vector are not associated with gene expression changes, unlike LPKO LCLs trans-complementated with EBNA-LP. A**. Heat map of core the identified 671 core YY1 binding sites in additional replicates for LPKO LCLs trans-complemented with wild type EBNA-LP or control vector. **B.** Volcano plot of YY1 binding sites enriched in LPKO LCLs trans-complemented with wild type EBNA-LP compared to LRM Mutant. Blue dots indicate significantly enriched sites (Log2 Fold Change above 0.7 and p value above 0.05). **C.** Enrichment of YY1 peaks enriched in LPKO LCLs trans-complemented with LRM Mutant compared to EBNA-LP in expression data sets from WT and LPKO LCL RNA-sequencing data. **D.** Volcano plot of YY1 binding sites enriched in LPKO LCLs trans-complemented with LRM mutant compared to vector. **E.** Enrichment of YY1 peaks enriched in LPKO LCLs LRM Mutant or **F.** Control vector in WT and LPKO RNA-seq data. **G.** Volcano plot of LPKO LCLs trans-complemented with EBNA-LP compared to control vector. **H.** Enrichment of YY1 binding sites enriched in LPKO LCLs trans-complemented with EBNA-LP or **I.** control vector in WT and LPKO LCLs RNA-seq data. NES = normalized enrichment score. FDR = false discovery rate q-value. Significance is defined as FDR less than 0.05.
(TIF)

## Acknowledgments

We thank Dr. Davide Maestri and Dr. Italo Tempera for their helpful advice on analysis of Cut&Run assays. We acknowledge Ashley P. Barry and Marc Leazer for generation of reagents. We thank the Duke University School of Medicine for the use of the Proteomics and Metabolomics Core Facility, which provided mass spectrometry services, and Erik Soderblom for assistance in interpreting the mass spectrometry data. We thank the Duke University School of Medicine for use of the Sequencing and Genomics Technologies Core Facility for sequencing of our RNA-sequencing and DNA-sequencing libraries. Flow Cytometry was performed in the Duke Cancer Institute Flow Cytometry Facility at Duke University, Durham, NC, which is supported by the NCI Cancer Center Support Grant (CCSG) award number P30CA014236. We also acknowledge the ENCODE Consortium and the lab of Richard Myers for generating the YY1 ChIP-Seq data.

## Author Contributions

**Conceptualization:** Jana M. Cable, Robert E. White, Micah A. Luftig.

**Data curation:** Jana M. Cable, Nicolás M. Reinoso-Vizcaino.

**Formal analysis:** Jana M. Cable.

**Funding acquisition:** Robert E. White, Micah A. Luftig.

**Investigation:** Jana M. Cable, Nicolás M. Reinoso-Vizcaino.

**Methodology:** Jana M. Cable.

**Project administration:** Robert E. White, Micah A. Luftig.

**Resources:** Robert E. White, Micah A. Luftig.

**Supervision:** Micah A. Luftig.

**Validation:** Jana M. Cable.

**Visualization:** Jana M. Cable.

**Writing – original draft:** Jana M. Cable.

**Writing – review & editing:** Jana M. Cable, Robert E. White, Micah A. Luftig.

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
