## [Decision Letter · Decision Letter 0]

22 Feb 2024

Dear Dr. Luftig,

Thank you very much for submitting your manuscript "Epstein-Barr virus protein EBNA-LP engages YY1 through leucine-rich motifs to promote naïve B cell transformation" for consideration at PLOS Pathogens. As with all papers reviewed by the journal, your manuscript was reviewed by members of the editorial board and by several independent reviewers. In light of the reviews (below this email), we would like to invite the resubmission of a significantly-revised version that takes into account the reviewers' comments.

Reviewers requested a few additional experiments to better support the claims.  Reviewer 1 suggest that you explore whether EBNA2 Chip assays overlap with the YY1/EBNA-LP.  Reviewer 2 raised concerns over the statistical robustness of some of the data presented in bar graphs for Figs. 1, 3 and 4.  This reviewer also requested some additional mechanistic data on the interaction and how mutation of the EBNA-LP leucine repeats affects YY1 function. Reviewer 3 raised concerns that the YY1 knock-out experiment did not show any EBV specific dependency. Reviewer 3 also requested experiments to examine the binding of EBNA-LP and YY1 to specific promoter sites and test if the leucine rich motif mutant LP alters YY1 binding (similar to reviewers 1 and 2). This would provide further evidence for the interaction and its potential function.   Additional minor concerns need to be addressed in the revision.

We cannot make any decision about publication until we have seen the revised manuscript and your response to the reviewers' comments. Your revised manuscript is also likely to be sent to reviewers for further evaluation.

Sincerely,

Paul M Lieberman

Academic Editor

PLOS Pathogens

Alison McBride

Section Editor

PLOS Pathogens

Michael Malim

Editor-in-Chief

PLOS Pathogens

orcid.org/0000-0002-7699-2064

Dear Micah

Your manuscript has been reviewed by 3 reviewers. Reviewers 2 and 3 had major revision requests for additional experiments. Reviewer 1 suggested that you explore whether EBNA2 Chip assays overlap with the YY1/EBNA-LP. Reviewer 2 raised concerns over the statistical robustness of some of the data presented in bar graphs for Figs. 1, 3 and 4. This reviewer also requested some additional mechanistic data on the interaction and how mutation of the EBNA-LP leucine repeats affects YY1 function. Reviewer 3 raised concerns that the YY1 knock-out experiment did not show any EBV specific dependency. Reviewer 3 also requested experiments to examine the binding of EBNA-LP and YY1 to specific promoter sites and test if the leucine rich motif mutant LP alters YY1 binding. 

Reviewer's Responses to Questions

**Part I - Summary**

Reviewer #1: This study by Cable et al establishes a novel assay for assessing the role of EBNA-LP for B cell transformation. The investigators then looked at the role of phosphorylation and leucine rich motifs for their contribution towards EBV-mediated B cell immortalization.

The strengths of this paper are the generation of a novel transcomplementation assay to assess the role of EBNA-LP during B cell transformation. An advantage of the assay is that cDNA clones of EBNA-LP can be used, which are more amenable for introduction of mutations than having to do this within the context of the viral genome. The work highlights the role of the LRM motifs within EBNA-LP for outgrowth of EBV-infected cells and a possible role for YY1 in this process.

Gene editing to knockout out YY1 showed its requirement for transformation, but how EBNA-LP’s interaction with YY1 plays a role in transformation remains somewhat hazy and speculative. The investigators suggest that YY1, perhaps through an interaction with EBNA-LP might play a role in gene regulation or other processes required for immortalization (e.g., regulation of apoptosis), although they state fairly emphatically in parts of the manuscript that this may have nothing to do with EBNA2 transcriptional regulatory functions. However, whether EBNA2 Chip assays overlap with the YY1/EBNA-LP maps is something that still might be worth exploring.

Reviewer #2: In this manuscript, the authors used transcomplementation to study the significance of different EBNALP domains in supporting EBV naïve B cell transformation. They found that the conserved serine residues are not required for LCL outgrowth and leucine repeat motifs are essential for naïve B cell transformation. The leucine repeats bind to YY1 and YY1 is essential for LCL growth.

Reviewer #3: This study examines the function of a key Epstein-Barr virus (EBV) encoded latent protein, EBNA leader protein (LP) in the transformation of naive B cells from cord blood, where it has been previously shown to play a key role. Studying the role of specific latent proteins such as EBNA LP in the transformation of B cells by EBV is important for our understanding of the cancer association of the virus and the mechanisms that underpin this. Understanding LP functions and dissecting its role in B cell transformation has been challenging and early information from transfection studies has provided conflicting information on the domains of EBNA LP required for its function and identified a transcriptional co-activation role (with EBNA2) which does not seem to be recapitulated in the context of whole virus infections.

This study sets up a complementation assay using wild-type and LPKO EBV to infect cord or adult B cells that can have been previously transfected with a vector expressing WTLP to complement the LPKO infection and allow examination of mutants. The system is well set up and validated and provides a good background to carry out mutant analysis. It suffers from the expected level of variation in infection and grow out efficiencies from donor to donor, but this does not diminish its usefulness.

The authors then examine the proposed role of phosphorylation of serine residues in LP in its B cell transformation function and show that mutation of none of the 3 key Serine residues (that they confirm can be phosphorylated by Mass Spec) affects cord blood transformation efficiency. This is an important result as phosphorylation was previously reported to be required in reporter assays for co-activation with EBNA2.

The authors then examine the potential role of Leucine rich motifs identified in LP that they propose may be involved in YY1 and PGC co-activator interactions based on YY1 co-localisation at LP binding sites by ChIP-seq and the presence of similar motifs as YY1 interaction motifs in PGC complexes. Mutants of two identified motifs did ablate cord blood transformation by LP. An interaction between LP and YY1 was also demonstrated by endogenous Co-IP in both directions that was lost with the double mutant. This therefore identifies a key new interaction partner that plays a role in EBNA LP dependent transfomation functions.

Overall the study is well executed and experiments are well designed and interpreted. There are a few further experiments that may solidify the findings.

**Part II – Major Issues: Key Experiments Required for Acceptance**

Reviewer #1: (No Response)

Reviewer #2: Major concerns:

1. The low consistency is a big problem for this manuscript. In figure 1, 3 and 4, there are 16 bar graphs, only 5 with P value less than 0.05. It is very difficult to draw solid conclusions when the statistics don’t support them. The author should reduce the variation.

2. The authors show an important interaction between EBNALP and YY1. However, the authors did not provide any mechanistic insights on why the interaction is important. The authors should at least provide some data on how does this mutant differ from wild-type in regulating gene expression and how do the leucine repeats affect YY1 function.

Reviewer #3: 1. The Cas9 knockout of YY1 is interesting and does show a dependency for B cell growth on YY1, but it is not clear from this experiment that this is an EBV specific dependency, given the broad role of YY1 in apoptosis, nor does it address any EBNA LP specific functional dependency. Could this experiment be done in the LCL complementation background to further investigate dependencies?

2. If EBNA-LP and YY1 overlap in their binding to chromatin and the interaction with YY1 requires the leucine rich motifs, it would be sensible to examine the binding of EBNA-LP and YY1 to specific promoter sites and show that the leucine rich motif mutant LP no longer binds/shows reduced binding. This would provide further evidence for the interaction and its potential function.

**Part III – Minor Issues: Editorial and Data Presentation Modifications**

Reviewer #1: 1. The investigators state that their infections were done to assure 100% infection efficiency. It would be useful to provide the particle to pfu ratio for their virus preparations to get a sense of “gene dosage” that the cells are getting. I can imagine that under biological conditions, naïve B cells may get infected by a single virus particle, where EBNA-LP for example, may be critical—even for EBNA2 transcriptional functions. However, if many of the cells in their current system are receiving 10’s or 100’s (or even 1000’s) of viral genomes, enhancing properties provided by EBNA-LP might be “masked” or even titration of inhibitors of transcription (or immortalization) might be occurring. In addition to gene dosage, “protein dosage” from a number of viral tegument proteins, such as BNRF1, may also negate the requirement of other viral factors during the immortalization process.

While the interpretations of this paper are within the boundaries of the experimental set-up, I believe some commentary is warranted to address this issue as some of their conclusions (role of phosphorylation or even EBNA2 transcriptional enhancement) might require modification if/or when different conditions are used.

2. The investigators should consider performing a simple experiment to see whether their LRM mutant affects EBNA2 transcriptional coactivation as it still may provide a surrogate assay for investigating EBNA-LP/YY1 mechanistic pathways.

3. Please address why the LRM mutant appears to migrate more slowly in the Immunoblots.

Reviewer #2: Minor concerns:

1. The authors should indicate if LCLs are established from transcomplementation.

2. Figure 1F, 2H, and 5A. The authors should use house keeping gene Western blot for loading control.

3. For Figure 1, 3, and 5, the Y axises are rTtomato+cells. Are these the total cell number from an experiment or pool from several wells? For optimum LCL growth, it is better to adjust the concentration to 1x105/ml. The concentration indicated in the figure might not be optimum for LCL growth.

4. The authors should consider pooling multiple transfections into one datapoint to improve the cell growth or use conditional medium.

5. 5. The authors should include memory B cells in the growth assay to give readers a more precise estimate of the transcomplementation efficiency.

Reviewer #3: Line 222-222 page 10 states that 'these results suggest mutation of the LRMs results in loss of an interaction between EBNA LP and a cellular protein'. I am not sure that this has been adequately demonstrated for this conclusion at this point, just that the mutation impacts on function. Suggest revising this.

PLOS authors have the option to publish the peer review history of their article (what does this mean?). If published, this will include your full peer review and any attached files.

Reviewer #1: No

Reviewer #2: No

Reviewer #3: No

Figure Files:

Data Requirements:

Reproducibility:

To enhance the reproducibility of your results, we recommend that you deposit your laboratory protocols in protocols.io, where a protocol can be assigned its own identifier (DOI) such that it can be cited independently in the futur

---

## [Decision Letter · Decision Letter 1]

30 Jun 2024

Dear Dr. Luftig,

We are pleased to inform you that your manuscript 'Epstein-Barr virus protein EBNA-LP engages YY1 through leucine-rich motifs to promote naïve B cell transformation' has been provisionally accepted for publication in PLOS Pathogens.

Best regards,

Paul M Lieberman

Academic Editor

PLOS Pathogens

Alison McBride

Section Editor

PLOS Pathogens

Michael Malim

Editor-in-Chief

PLOS Pathogens

orcid.org/0000-0002-7699-2064

Please respond to comments from reviewer 2 relating to statistical tests.

Reviewer Comments (if any, and for reference):

Reviewer's Responses to Questions

**Part I - Summary**

Reviewer #1: (No Response)

Reviewer #2: (No Response)

Reviewer #3: The authors have adequately addressed the points I raised previously with additional data, further analysis or text changes

**Part II – Major Issues: Key Experiments Required for Acceptance**

Reviewer #1: (No Response)

Reviewer #2: The authors included new statistical analyses for the transformation assay. However, the Fisher’s exact test results are disconnected from other tests. For example, in Figure 4, none of the individual tests were significant, yet the Fisher’s exact test was highly significant. Is Fisher’s exact test the right assay here? I still believe the authors should increase the sample size to reduce the inconsistency.

Reviewer #3: (No Response)

**Part III – Minor Issues: Editorial and Data Presentation Modifications**

Reviewer #1: (No Response)

Reviewer #2: (No Response)

Reviewer #3: (No Response)

PLOS authors have the option to publish the peer review history of their article (what does this mean?). If published, this will include your full peer review and any attached files.

Reviewer #1: No

Reviewer #2: No

Reviewer #3: No

---

## [Editor Report · Acceptance letter]

23 Jul 2024

Dear Dr. Luftig,

We are delighted to inform you that your manuscript, "Epstein-Barr virus protein EBNA-LP engages YY1 through leucine-rich motifs to promote naïve B cell transformation," has been formally accepted for publication in PLOS Pathogens.

Best regards,

Michael Malim

Editor-in-Chief

PLOS Pathogens

orcid.org/0000-0002-7699-2064